# Social determinants and risk factors associated with non-communicable diseases among urban population in Nepal: A comparative study of poor, middle and rich wealth categories of urban population using STEPS survey

**Sampurna Kakchapati**[1]*, **Raju Neupane**[1], **Kriti Sagar Baral**[1], **Grishu Shrestha**[1], **Deepak Joshi**[1], **Bryony Dawkins**[2], **Tim Ensor**[3], **Helen Elsey**[4], **Sushil Chandra Baral**[1]

**1** HERD International, Lalitpur, Nepal, **2** Academic Unit of Health Economics, Leeds Institute of Health Sciences, University of Leeds, Leeds, United Kingdom, **3** Nuffield Centre for International Health and Development, Leeds Institute of Health Sciences, University of Leeds, Leeds, United Kingdom, **4** Hull York Medical School, Department of Health Sciences, University of York, Heslington, United Kingdom

* kck_sampurna@yahoo.com, sampurna.kakchapati@herdint.com

## Abstract

### Background

Non-communicable diseases (NCDs) are significant public health concern globally, and the burden is disproportionately high among urban populations. This study aims to compare the social determinants, NCD risk, and NCD prevalence among different wealth categories and to determine the factors associated with hypertension, obesity, and diabetes among the urban population of Nepal.

### Methods

This study used urban population data from cross-sectional STEP wise approach to NCD risk factor surveillance (STEPS) survey of 2019, resulting in a sample of 3460 individuals of 15-69 years for inclusion in the analysis. We used bivariate analysis to compare the social determinants, NCD risk and NCD prevalence among urban poor, urban middle and urban rich and multivariate logistic regression to determine the association between social determinants, NCD risks and obesity, hypertension and diabetes among urban population.

### Results

The study found significant differences in hypertension, obesity and diabetes by gender, ethnicity, education, employment, smoking habits, and cholesterol levels between the three wealth groups. Among the urban poor, low education, unemployment and smoking habits were more prevalent, while high cholesterol was more prevalent among the urban rich.

**Data availability statement:** All relevant data are within the article and its Supporting Information files.

**Funding:** The author(s) received no specific funding for this work.

**Competing interests:** The authors have declared that no competing interests exist.

The significant factors associated with overweight and obesity after Bonferroni correction included Hilly region with higher odds of overweight (AOR=2.33, 95% CI=1.45-3.75,). In contrast, being from Karnali (AOR= 0.36, 95% CI=0.22-0.58) and Sudurpaschim (AOR=0.42, 95% CI=0.26-0.66) provinces were associated with lower odds of overweight and cholesterol, while cholesterol was associated with higher odds of obesity (AOR=1.01, 95% CI=1.01-1.02). Disadvantaged janajatis had the lower odds of overweight (AOR = 0.52, 95% CI = 0.36-0.78). Factors that remained significantly associated with hypertension and pre-hypertension after Bonferroni correction included: age, with higher odds of hypertension (AOR=1.03, 95% CI=1.02-1.04); men, who had higher odds of both pre-hypertension (AOR=1.68, 95% CI=1.19-2.36) and hypertension (AOR=2.23, 95% CI=1.56-3.47). Being obese (AOR = 5.12, 95% CI = 2.95-8.87, p = 0.001) and overweight (AOR = 1.69, 95% CI = 1.19-2.39, p = 0.003) were significantly associated with hypertension. Similarly, urban population residing in the hilly region had higher odds of diabetes (AOR=6.44, 95% CI=3.31-11.10) compared to the mountain region; those living in the Tarai region had higher odds of pre-diabetes (AOR=5.07, 95% CI=2.44-10.5) and diabetes (AOR=5.96, 95% CI=3.12-19.86). Respondents with high cholesterol higher odds of both pre-hypertension (AOR=1.00, 95% CI=1.00–1.02) and hypertension (AOR=1.03, 95% CI=1.02-1.04), pre-diabetics (AOR=1.00, 95% CI=1.00–1.02) and diabetics (AOR=1.03, 95% CI=1.02-1.04).

## Conclusion

The findings indicate significant disparities in education, employment, and lifestyle habits across wealth groups; urban poor lacked education and employment. Factors such as ecological region, province, gender and age were associated with an increased risk of various health conditions such as being overweight, pre-hypertension, pre-diabetes, and diabetes. Improved health outcomes among urban populations interventions targeting increased access to education, additional investment in specific areas where outcomes are worst, and interventions to improve equitable access to healthcare are needed.

## Introduction

Non-communicable diseases (NCDs) are now the leading global causes of death. Diseases such as diabetes, cardiovascular diseases, cancer and chronic respiratory diseases are collectively responsible for 74% of worldwide deaths [1]. A total of 41 million people die each year because of NCDs, 17 million of these are under 70 years of age and more than 85% of these premature deaths are in low- and middle-income countries [2]. This disproportionate distribution of deaths in low- and middle-income countries is fundamentally an issue of equity and social justice [3]. In these countries, the conditions in which people are born, grow, work, live and age referred to as the social determinants of health play a significant role in driving NCD and NCD

risk factors [3,4]. These determinants are shaped by equal systems of wealth, powered and resources that stratify populations by socioeconomic status, gender, ethnicity, and disability, leading to unequal health outcomes [5,6,7]. As a result, socio-economic factors and behavioral risk factors, such as poor diet, tobacco use, physical inactivity and harmful use of alcohol are closely associated with morbidity and mortality for NCDs [3,4].

In Nepal, the impact of urbanisation, economic development, and changing lifestyles has created both opportunities and challenges in the thriving urban environment [3,6,8]. As people migrate to urban areas seeking for better economic opportunities, cities often experience strained infrastructure and health services [9,10]. Vulnerable populations, including low-income families, rural-to-urban migrants, and residents of informal settlements, are particularly affected. These groups frequently face limited access to healthcare, poor living conditions, and environments that promote unhealthy lifestyles [10,11]. As a result, they are at increased risk of developing non-communicable diseases (NCDs), such as hypertension, diabetes, and cardiovascular diseases, due to factors like inadequate nutrition, reduced physical activity, and insufficient healthcare access [4,5,7].

The dynamics of urban health in Nepal are associated with a complex and interlinked range of wider determinants [10,11], as conceptualized by the Commission on Social Determinants of Health (CSDH) [7]. Urban residents, particularly those in low-income and informal settlements, often experience financial limitations that restrict their ability to purchase nutritious food, access quality healthcare, and maintain healthy lifestyles [12,13]. These constraints are driven by high living costs in cities, including housing, transportation, and utilities, leaving little disposable income for health-promoting activities [8,10]. As a result, residents may rely on cheaper, unhealthy food options, forgo preventive healthcare services, and struggle to engage in regular physical activity due to a lack of safe recreational spaces or time constraints associated with long working hours [10,12,13]. Furthermore, healthcare facilities in these areas are often overcrowded or of poor quality, making timely and effective medical care inaccessible [6,9,11]. Psychosocial factors, including heightened stress levels and inadequate mental health support, particularly among informal settlement dwellers, has lead to a surge in mental health disorders [5,13]. The co-influence of socio-economic factors, including gender and wealth disparities, further magnifies health inequities, creating a complex interplay between social determinants and health outcomes [3,4,10].

Moreover, the urban populations remain a neglected population and evidence and interventions regarding social determinants and risk factors of NCDs among urban population are scarce in many low and middle income countries, including Nepal [3,4,7]. Within cities, there is a lack of studies comparing different wealth strata. This is in part due to the limited data available from informal settlements and urban poor households, who may be missed in survey sampling frames and even within the census in the country [13]. Understanding the intricate relationship between social determinants and NCDs among the urban population in Nepal is essential for several reasons [11,13,14]. It provides critical insights into the underlying structural factors that drive health disparities within this demographic. In this context, this study utilized the data of the 2019 STEPS survey to compare the prevalence of risk factors across wealth quintiles and provide analysis of the factors associated with NCDs within the urban population [15]. The study hypothesis that social determinants, such as wealth quintile and other socio-economic factors, are significantly associated with the prevalence of NCD risk factors among the urban population in Nepal. This study will contribute valuable insights for the prevention, control and reduction of NCDs by providing relevant information on social determinants and risk factors of NCDs among urban population.

## Methods

### Study setting

Nepal is a landlocked country in South Asia between India and China, geographically divided into three main regions: the sparsely populated mountainous region, the densely populated hilly region with a mix of urban and rural settlements, and the agriculturally significant Tarai. Nepal's urbanization rate has shown a gradual increase, with 66% of the population residing in cities in 2021, up from 62.9% in 2011 and 13.9% in 2001. The significant rise between 2001 and 2011 reflects a period of accelerated urban growth, which has slowed in the subsequent decade [16]. Significant internal migration

occurs, particularly from the Tarai and mountainous regions to urban centers in the hilly region, notably Pokhara and Kathmandu. An estimated 70 percent of Nepalese migrate to the urban areas for employment and education [16]. Nepal is divided into 7 provinces and 753 local levels. The highest urban population concentration is in Madhesh province (19.9%), followed by Bagmati province (14.6%) and the lowest in Karnali province (1.2%), which shows an imbalance in the distribution of the urban population [16].

## Study design

This study utilized data from the STEP wise approach to NCD risk factor surveillance (STEPS) survey, a comprehensive cross-sectional survey conducted by World Health Organization (WHO) and Nepal Health Research Council (NHRC) to monitor and evaluate NCD risk factors and related health behaviors [15]. The survey was designed to collect data on the prevalence and distribution of risk factors for NCDs, such as tobacco use, alcohol consumption, physical inactivity, and unhealthy diet. The survey employed a multistage, stratified sampling design to ensure representative coverage of the target population. Data were collected through face-to-face interviews and standardized physical measurements conducted by trained field personnel. The survey instrument included structured questionnaires covering a range of demographic, socioeconomic, lifestyle, and health-related variables. Additionally, physical measurements such as blood pressure, anthropometric indices, and biochemical parameters were collected following standardized protocols.

## Participants

The STEPS Survey 2019 was designed to be representative of the national population and was carried out between February and May 2019. The survey population included the adult population from 15 to 69 years. The survey used a multistage cluster design: At the first stage, 259 wards were selected from the whole country in which there were total 37 primary sampling units (PSUs) in each province. In the second stage, 25 households were selected from each of the PSU out of the listed sampling frame of households in each ward using systematic random sampling [14,15]. From each of the selected household, one adult member was sampled randomly for participation in the survey using the android tablet. The STEP dataset included about 6,475 eligible individuals, from which we included 3460 individuals from urban areas in our analysis. The selection of 3,460 participants was based on a subset of individuals who were classified as residing in urban areas from the total population of 6,475 participants in the original dataset. The selection was guided by predefined criteria for urban residence, ensuring that only those participants who live in metropolitan cities, sub-metropolitan cities, municipalities were considered as urban population and participants from rural municipalizes we considered as rural population.

## Merging of Wealth Quintiles

In this study, wealth quintiles, initially categorized into five groups, were merged into three broader categories—Urban Poor, Urban Middle, and Urban Rich—due to significant disparities in sample sizes across the quintiles. The first two quintiles were combined into the Urban Poor category, the fourth and fifth quintiles into the Urban Rich category, while the third quintile was kept separate as the Urban Middle category for analytical clarity and comparison. This merging was necessary to address sample size imbalances, ensuring each category had an adequate sample for statistical analysis. The focus of the analysis was on the wealth extremes to better understand the social determinants and risk factors for NCDs in the richest and poorest urban populations. The middle quintile was preserved to facilitate comparisons. The distribution of wealth quintile in urban population is presented in S1 Table.

Table 1 shows the list of variables and its definitions used in the study.

## Study variables

The primary outcomes of interest in this study were three common NCDs: obesity, hypertension and diabetes. The independent variables selected from the study was adapted from WHO Commission on Social Determinants of Health

**Table 1. Variable definitions.**

| Variables | Definitions |
|---|---|
| Ethnicity groups | Defined using categories in the Nepal Demographic Health Survey as Upper caste Groups, Advantaged Janajatis, Disadvantaged groups, Dalits and Religious Minorities |
| Ecological region | Ecological region in Nepal is divided into Mountain, Hilly and Tarai regions |
| Province | Provinces of Nepal is categorized as Koshi, Madhesh, Bagmati, Gandaki, Lumbini, Karnali and Sudurpashchim |
| Education | Education Status is categorize into four levels as No formal education includes individuals who have never attended any form of school, either formal or informal. Primary education includes those who attended or completed grades 1 to 5 but did not continue to secondary education. Secondary education includes middle school and high school, covering grades 6–12. University education who have attended post-secondary or higher education including university, college, or other tertiary-level institutions. |
| Wealth quintile | Wealth quintiles was classified into 5 category- Lowest, Second, Middle, Fourth and Highest Wealth Quintiles. |
| Rich Category | Highest and fourth wealth quintiles were combined to form the Rich category. |
| Middle Category | Middle wealth quintile was used in the analysis as the middle category. |
| Poor Category | Lowest and Second wealth quintiles were combined to form the Poor category. |
| Current smoker | Participants who had smoked in the past 30 days were considered as current smokers for this survey. |
| Harmful alcohol consumption | Harmful use means consumption of greater than or equal to 60 gm of pure alcohol on an average day in the past 30 days. The term "average day" refers to the typical or usual amount of alcohol consumed per day by an individual over a specific period in the past 30 days. |
| Sufficient fruit and vegetable consumption | Participants who ate five or more servings of fruits and vegetables per day |
| Sufficient physical activity | Participants who participated in more than or equal to 150 minutes of moderate intensity (600 METs) physical activities per week |
| 24 hours Salt Consumption | Participants who had consumed salts within 24 hours was estimated using the INTERSALT Southern European equation to measure 24 hour mean salt intake [17]. |
| Obesity | Participants with a BMI greater than or equal to $30 \, kg/m^2$, were classified as being obese. |
| Overweight | Participants with a BMI greater than or equal to $25 \, kg/m^2$, were classified as being overweight. |
| Pre-hypertension | Participants were classified as having Pre-Hypertension if the average 2nd and 3rd measurement of systolic BP was greater to 120 mmHg but less than 140 mmHg, or the average diastolic BP was greater than 80 mmHg but less than 90 mmHg, or if they reported to be taking antihypertensive medication [18]*. |
| Hypertension | Participants were classified as having Hypertension if the average 2nd and 3rd measurement of systolic BP was greater or equal to 140 mmHg, or the average diastolic BP was greater than 90 mmHg, or if they reported to be taking antihypertensive medication. |
| Pre-diabetes | Participants with a fasting blood sugar greater than 99 mg/dl and less than 126 mg/dl |
| Diabetes | Participants with a fasting blood sugar greater or equal to 126 mg/dl, or those currently taking medications to lower blood sugar, were considered to be Diabetics |
| Urban Poor | Participants from the urban population belonging to the poor wealth category, which includes both the poorest quintile (1014 participants) and the second quintile (680 participants), resulting in a combined sample size of 1694 participants. |
| Urban Middle | Participants from the urban population belonging to the middle wealth category, which includes both the third quintile (590 participants) |
| Urban Rich | Participants from the urban population belonging to the rich wealth category, which includes both the fourth quintile (550 participants) and the fifth quintile (626 participants), resulting in a combined sample size of 1176 participants. |

*Definitions are based on the American College of Cardiology (ACC) and American Heart Association (AHA) guidelines (2020).

(CSDH). The structural determinants of health include social class (Gender, Marital Status, Ethnicity, Ecological belt and Province) along with socio-economic position (Education, Occupation and Wealth Quintile). Likewise, the intermediary determinants of health include behavior and biological factors such as alcohol and smoking use, fruit and vegetable consumption, 24 hours salt consumption, sufficient physical activity, age and cholesterol level.

Fig 1 depicts the he conceptual framework showing the structural determinants, socio economic position, intermediary determinants and outcomes. The framework illustrates how the structural determinants such as social class along with

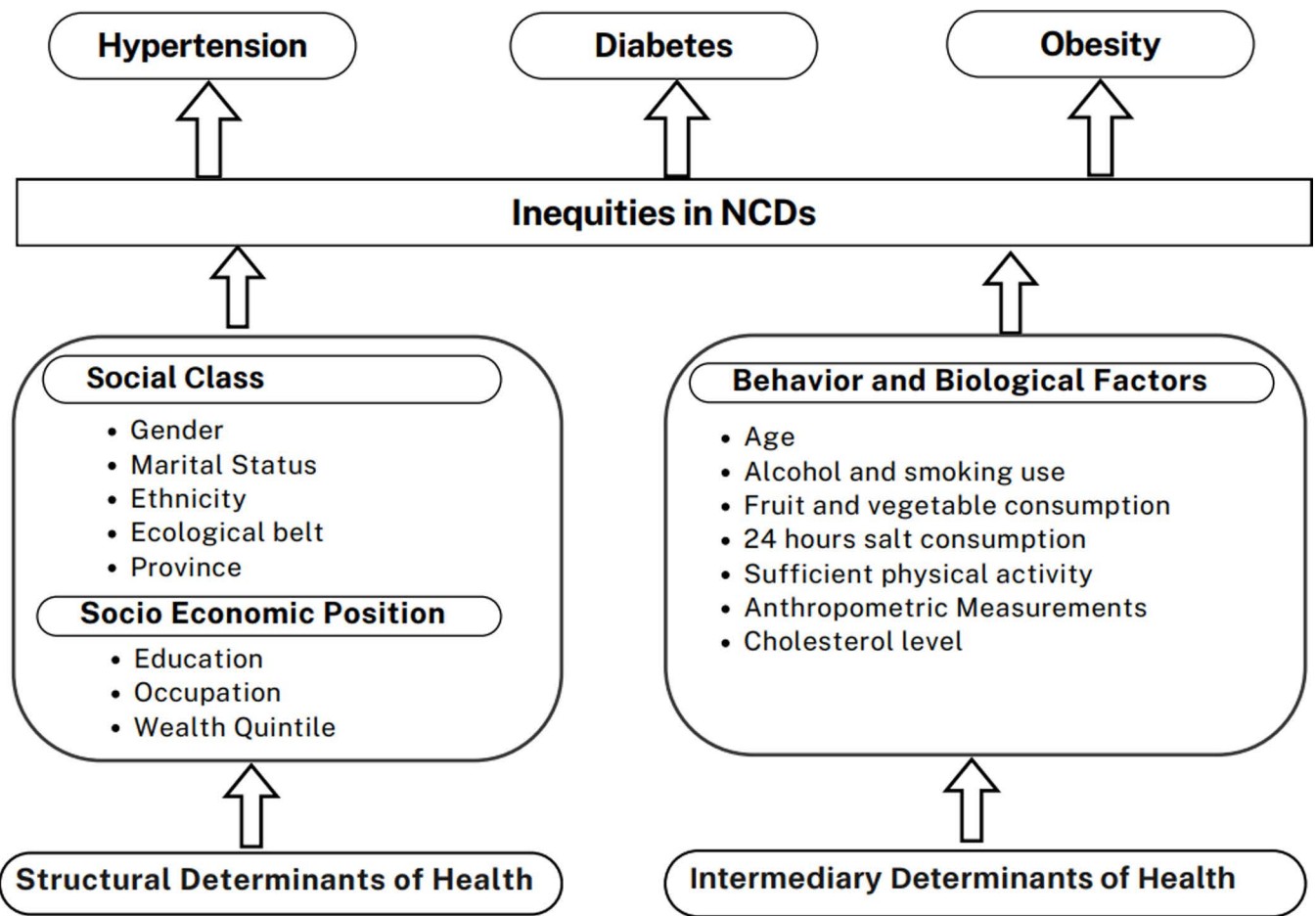

**Fig 1. Conceptual Framework for analysis of NCDs and its relationship with social determinants of health.**

socio-economic position influence the inequalities of NCDs such as cardiovascular diseases and diabetes. Intermediary determinants like behavior and biological factors directly impact health outcomes, particularly NCDs such as cardiovascular diseases and diabetes.

## Statistical methods

Bivariate analyses were conducted to assess the unadjusted associations between the independent variables among urban poor, urban middle and urban rich using chi-square tests for categorical variables and t-tests or ANOVA for continuous variables, as appropriate. We used R version 4.2.0 for statistical analysis and used "survey" package 15 and performed a weighted analysis to account for the complex survey design of 2020 STEP. Weighted analysis ensures that the results are representative of the population by adjusting for unequal probabilities of selection, non-response, and other factors that might bias the results if ignored. We used univariate and multivariate weighted logistic regression analysis to determine the association of the NCDs with independent variables including age, gender, ecological region, province, ethnicity, marital status, education, occupation, smoking, alcohol consumption, physical activity, salt intake. Initially, explanatory determinants were included in the model one at a time to examine their univariate relationship with the outcome and variables that were significant in the univariate analysis were fitted in the multivariate analysis. Variables that were

statistically significant in the univariate analysis were then considered for inclusion in the multivariate analysis Multivariate weighted logistic regression models were then used to identify the most important determinants for each outcome. In all analyses, Bonferroni correction was applied to adjust for multiple comparisons. Weights in the multivariate analysis allow for more accurate population-level estimates, accounting for the complex survey design and providing unbiased parameter estimates. A p-value of less than 0.05 was used to define statistical significance. Adjusted odds ratios (AORs) as well as their 95% confidence intervals (95% CI) were used to depict the independent relationship between predictors and dependent variables.

## Ethics statement

The data of STEP survey were approved by (Nepal Health Research Council (NHRC)). Ethical approval to conduct this survey was granted from the Ethical Review Board (ERB) of the NHRC, Government of Nepal (Registration number 293/2018). The data were made publicly available after maintaining the anonymity. All data used in this study were fully anonymized before being accessed and identifiers such as patient names, addresses, and contact details were removed to protect patient confidentiality.

## Results

Table 2 shows the comparison of socio demographic factors and risk behaviors that include, behavioral factors, biological factors and NCD among poor, middle and rich wealth categories of urban population. The merging of wealth quintiles into three categories—Urban Poor, Urban Middle, and Urban Rich—is reflected in the analysis of wealth disparities in relation to non-communicable diseases (NCDs). This merging process addressed sample size imbalances and provided a clearer focus on the wealth extremes, while maintaining the middle quintile for comparison. Slightly less than half of the urban population (49%) are in the poor wealth category whereas about one third of urban population are in rich category (34%). There was a significant association among education status, occupation status, current use of tobacco products and cholesterol level within the three wealth categories of urban population (p < 0.05). Urban population with higher levels of education are more prevalent in the rich category, whereas those with little or no formal education are more likely to be poor. Employment is higher in the middle and rich category, while homemakers are more predominant in the poor category. Smokers are more prevalent in the rich category compared to middle and poor category. There were notable variations in total cholesterol across different economic strata and the urban rich category had higher cholesterol levels compared to middle and poor categories.

Table 3 shows the factors associated with being overweight and obese in multivariate logistic regression among urban population. Disadvantaged ethnic groups had lower odds of being overweight (AOR=0.52, 95% CI =0.36- 0.78, p=0.001) and lower odds of being obese (AOR=0.45, 95% CI =0.22-0.92, p=0.028) compared to advantaged ethnic groups. Despite the odds ratio indicating lower obesity odds, the Bonferroni-adjusted p-value indicates no statistical significance after multiple testing correction. Urban Population in the Hilly region (AOR=2.33, 95% CI = 1.45-3.75, p = 0.001) had higher odds of being overweight compared to the mountain region. This association remains significant even after Bonferroni correction (Bonferroni p = 0.011). Likewise, urban population from Bagmati province had higher odds of obesity (AOR=2.36, 95% CI = 1.45-3.75, p = 0.001) and overweight (AOR = 1.82, 95% CI = 1.18- 2.82, p = 0.007) compared to Koshi province, however, after Bonferroni correction (Bonferroni p = 0.15), this result is not statistically significant. Urban population in Karnali province (AOR = 0.36, 95% CI = 0.22-0.58, p = 0.001 and Sudurpaschim province (AOR=0.42, 95%CI=0.26-0.66, p=0.001) had lower odds of being overweight compared to Koshi province and this result is highly significant before and after Bonferroni correction. Ethnicity was significant associated with overweight and obesity as disadvantaged janajatis had the lower odds of overweight (AOR = 0.52, 95% CI = 0.36-0.78, p = 0.001) and obesity compared to advantaged Janajatis and these results was found to be significant for overweight after Bonferroni correction. Urban population who had ever been married had higher odds of being overweight (AOR = 2.11, 95% CI = 1.07- 4.13, p = 0.03) compared to

**Table 2. Comparison of social determinants, NCD risk behaviors and NCD among urban poor, urban middle and urban rich.**

| Characteristics | Wealth Categories in Urban Population | | | | P value |
| --- | --- | --- | --- | --- | --- |
| | Urban Poor (n=1694, 49%) | Urban Middle (n=590, 17%) | Urban Rich (n=1176, 34%) | Total (N=3460) | |
| **Socio Demographic Factors** | | | | | |
| Age* | | | | | 0.229 |
| Median (IQR) | 38 (30,51) | 37 (28,50.8) | 38 (29,49) | 38 (29,50) | |
| Gender | | | | | 0.063 |
| Female | 1120 (66.1) | 369 (62.5) | 731 (62.2) | 2220 (64.2) | |
| Male | 574 (33.9) | 221 (37.5) | 445 (37.8) | 1240 (35.8) | |
| Ecological Belt | | | | | 0.352 |
| Mountain | 164 (9.7) | 51 (8.6) | 95 (8.1) | 310 (9) | |
| Hilly | 710 (41.9) | 240 (40.7) | 471 (40.1) | 1421 (41.1) | |
| Tarai | 820 (48.4) | 299 (50.7) | 610 (51.9) | 1729 (50) | |
| Province | | | | | 0.121 |
| Koshi | 246 (14.5) | 68 (11.5) | 161 (13.7) | 475 (13.7) | |
| Madhesh | 267 (15.8) | 102 (17.3) | 214 (18.2) | 583 (16.8) | |
| Bagmati | 279 (16.5) | 84 (14.2) | 208 (17.7) | 571 (16.5) | |
| Gandaki | 231 (13.6) | 98 (16.6) | 158 (13.4) | 487 (14.1) | |
| Lumbini | 210 (12.4) | 68 (11.5) | 148 (12.6) | 426 (12.3) | |
| Karnali | 216 (12.8) | 69 (11.7) | 123 (10.5) | 408 (11.8) | |
| Sudurpashchim | 245 (14.5) | 101 (17.1) | 164 (13.9) | 510 (14.7) | |
| Education Status | | | | | **0.029** |
| No formal education | 807 (47.6) | 270 (45.8) | 525 (44.6) | 1602 (46.3) | |
| Primary | 317 (18.7) | 123 (20.8) | 201 (17.1) | 641 (18.5) | |
| Secondary | 493 (29.1) | 177 (30) | 408 (34.7) | 1078 (31.2) | |
| University | 77 (4.5) | 20 (3.4) | 42 (3.6) | 139 (4) | |
| Ethnicity | | | | | 0.178 |
| Advantaged Janajatis | 277 (16.4) | 94 (15.9) | 177 (15.1) | 548 (15.8) | |
| Dalits | 225 (13.3) | 97 (16.4) | 149 (12.7) | 471 (13.6) | |
| Disadvantaged groups | 472 (27.9) | 157 (26.6) | 373 (31.7) | 1002 (29) | |
| Religious Minorities | 65 (3.8) | 19 (3.2) | 37 (3.1) | 121 (3.5) | |
| Upper caste Groups | 655 (38.7) | 223 (37.8) | 440 (37.4) | 1318 (38.1) | |
| Marital Status | | | | | 0.136 |
| Unmarried | 153 (9) | 70 (11.9) | 117 (9.9) | 340 (9.8) | |
| Ever Married | 1541 (91) | 520 (88.1) | 1059 (90.1) | 3120 (90.2) | |
| Occupation Status | | | | | **0.018** |
| Unemployed | 107 (6.3) | 26 (4.4) | 50 (4.3) | 183 (5.3) | |
| Students | 117 (6.9) | 46 (7.8) | 86 (7.3) | 249 (7.2) | |
| Home maker | 935 (55.3) | 314 (53.3) | 603 (51.3) | 1852 (53.6) | |
| Employed | 427 (25.2) | 153 (26) | 335 (28.5) | 915 (26.5) | |
| Others | 106 (6.3) | 50 (8.5) | 102 (8.7) | 258 (7.5) | |
| **Behavioral Factors** | | | | | |
| Current Smoking | | | | | **0.003** |
| No | 1358 (80.2) | 496 (84.1) | 998 (84.9) | 2852 (82.4) | |
| Yes | 336 (19.8) | 94 (15.9) | 178 (15.1) | 608 (17.6) | |
| Alcohol Consumption | | | | | 0.694 |
| No | 1233 (72.8) | 440 (74.6) | 859 (73) | 2532 (73.2) | |
| Yes | 461 (27.2) | 150 (25.4) | 317 (27) | 928 (26.8) | |

*(Continued)*

**Table 2.** (Continued)

| Characteristics | Wealth Categories in Urban Population | | | | P value |
|---|---|---|---|---|---|
| | Urban Poor (n=1694, 49%) | Urban Middle (n=590, 17%) | Urban Rich (n=1176, 34%) | Total (N=3460) | |
| Sufficient fruit and vegetable consumption | | | | | 0.263 |
| Insufficient | 1433 (84.6) | 515 (87.3) | 1008 (85.7) | 2956 (85.4) | |
| Sufficient | 261 (15.4) | 75 (12.7) | 168 (14.3) | 504 (14.6) | |
| Sufficient physical activity | | | | | 0.936 |
| Insufficient | 289 (17.1) | 97 (16.4) | 197 (16.8) | 583 (16.8) | |
| Sufficient | 1405 (82.9) | 493 (83.6) | 979 (83.2) | 2877 (83.2) | |
| 24 hours Salt Consumption | | | | | 0.985 |
| No | 355 (21) | 122 (20.7) | 247 (21) | 724 (20.9) | |
| Yes | 1339 (79) | 468 (79.3) | 928 (79) | 2735 (79.1) | |
| **Biological Factors** | | | | | |
| Total Cholesterol* | | | | | **0.004** |
| Median (IQR) | 139 (114,169) | 134 (114.2,167) | 144 (118,171) | 140 (115,170) | |
| BMI Categories | | | | | 0.366 |
| Normal | 983 (58.9) | 338 (58.5) | 660 (56.7) | 1981 (58.1) | |
| Obesity | 116 (7) | 36 (6.2) | 72 (6.2) | 224 (6.6) | |
| Overweight | 410 (24.6) | 153 (26.5) | 333 (28.6) | 896 (26.3) | |
| Underweight | 160 (9.6) | 51 (8.8) | 99 (8.5) | 310 (9.1) | |
| Health Outcomes | | | | | |
| Diabetes | | | | | 0.337 |
| Normal | 132 (7.8) | 34 (5.8) | 83 (7.1) | 249 (7.2) | |
| Pre-diabetic | 1205 (71.1) | 419 (71) | 820 (69.7) | 2444 (70.6) | |
| Diabetic | 357 (21.1) | 137 (23.2) | 273 (23.2) | 767 (22.2) | |
| Hypertension | | | | | 0.554 |
| Normal | 455 (26.9) | 158 (26.8) | 303 (25.8) | 916 (26.5) | |
| Pre-Hypertension | 892 (52.7) | 300 (50.8) | 602 (51.2) | 1794 (51.8) | |
| Hypertension | 347 (20.5) | 132 (22.4) | 271 (23) | 750 (21.7) | |

**\*t-test**

urban population who had never married. After Bonferroni correction, this association is no longer statistically significant (Bonferroni p = 0.63). Urban population with high cholesterol had higher odds of being overweight (AOR = 1.01, 95% CI = 1.00–1.02, p < 0.001) and higher odds of being obese (AOR = 1.01, 95% CI = 1.01–1.02, p < 0.001) compared to those with normal cholesterol levels. These results remain statistically significant for obesity after Bonferroni correction (Bonferroni p = 0.0004).

Table 4 depicts the factors associated with pre-hypertension and hypertension in multivariate logistic regression among urban population. Age is found to be significantly associated with hypertension (AOR= 1.03,95%CI 1.02-1.04, p<0.001) and this association remains significant even after Bonferroni correction (Bonferroni p <0.001). In addition, men have significantly higher odds of both pre-hypertension (AOR = 1.68, 95% CI = 1.19-2.36, p = 0.003) and hypertension (AOR = 2.23, 95% CI = 1.56-3.47, p<0.001) compared to women and these results remain statistically significant even after Bonferroni correction. Similarly, alcohol consumption (AOR = 1.58, 95% CI = 1.08-2.31, p = 0.019) were associated with hypertension, this association remains insignificant even after Bonferroni correction (Bonferroni p = 0.321). Likewise, being in upper caste had the higher odds of prehypertension (AOR = 1.78, 95% CI = 1.13-2.79, p = 0.013). Furthermore, occupation wise, students had the lower odds of hypertension (AOR=0.25, 95%CI 0.09-0.68, p=0.007) while

**Table 3. Factors associated with Overweight and Obesity in univariate and multivariate logistic regression among Urban Population.**

| Characteristics | Overweight Vs Normal | | | | | Obesity Vs Normal | | | | |
|---|---|---|---|---|---|---|---|---|---|---|
| | Unadjusted | | Adjusted | | Bonferroni p-value | Unadjusted | | Adjusted | | Bonferroni p-value |
| | OR (95% CI) | p-value | OR (95% CI) | p-value | | OR (95% CI) | p-value | OR (95% CI) | p-value | |
| **Age*** | 1.02(1.01, 1.03) | **<0.001** | 1(0.99, 1.01) | 0.86 | 1.0 | 1.03(1.02, 1.04) | **<0.001** | 1.01(1.00, 1.02) | 0.18 | 1.0 |
| **Sex** | | | | | | | | | | |
| Women | Ref | | | | | Ref | | | | |
| Men | 1.02(0.78, 1.33) | 0.88 | | | | 0.66(0.43, 1.02) | 0.06 | | | |
| **Ecological Region** | | | | | | | | | | |
| Mountain | Ref | | Ref | | | Ref | | | | |
| Hilly | 2.49(1.30, 4.79) | **0.01** | 2.33(1.45, 3.75) | **<0.001** | **0.011** | 1.74(0.59, 5.13) | 0.32 | | | |
| Tarai | 1.22(0.65, 2.26) | 0.53 | 1.57(0.96, 2.56) | 0.07 | 1.0 | 0.87(0.30, 2.51) | 0.79 | | | |
| **Province** | | | | | | | | | | |
| Koshi | Ref | | Ref | | | Ref | | Ref | | |
| Madhesh | 0.89(0.47, 1.66) | 0.7 | 0.96(0.52, 1.78) | 0.91 | 1.0 | 0.72(0.30, 1.71) | 0.45 | 0.73(0.31, 1.69) | 0.46 | 1.0 |
| Bagmati | 2.03(1.11, 3.70) | **0.02** | 1.82(1.18, 2.82) | **0.007** | 0.15 | 2.36(1.11, 5.04) | **0.03** | 2.36(1.07, 5.19) | **0.033** | 0.63 |
| Gandaki | 1.25(0.69, 2.28) | 0.46 | 1.10(0.70, 1.72) | 0.68 | 1.0 | 1.62(0.72, 3.63) | 0.24 | 1.46(0.61, 3.47) | 0.39 | 1.0 |
| Lumbini | 0.76(0.41, 1.39) | 0.36 | 0.86(0.43, 1.74) | 0.68 | 1.0 | 0.85(0.32, 2.26) | 0.74 | 0.78(0.27, 2.23) | 0.64 | 1.0 |
| Karnali | 0.42(0.23, 0.76) | **<0.001** | 0.36(0.22, 0.58) | **<0.001** | **0.0009** | 0.4(0.13, 1.31) | 0.13 | 0.51(0.16, 1.69) | 0.27 | 1.0 |
| Sudurpashchim | 0.32(0.19, 0.55) | **<0.001** | 0.42(0.26, 0.66) | **<0.001** | **0.004** | 0.32(0.12, 0.84) | **0.02** | 0.4(0.14, 1.14) | 0.087 | 1.0 |
| **Education** | | | | | | | | | | |
| No formal education | Ref | | | | | Ref | | | | |
| Primary | 0.95(0.66, 1.36) | 0.78 | | | | 1(0.61, 1.65) | 0.99 | | | |
| Secondary | 0.95(0.70, 1.28) | 0.74 | | | | 0.69(0.39, 1.20) | 0.18 | | | |
| University | 1.46(0.84, 2.55) | 0.18 | | | | 1.16(0.45, 2.99) | 0.76 | | | |
| **Wealth Quintile** | | | | | | | | | | |
| Lower Index | Ref | | | | | Ref | | | | |
| Middle Index | 0.91(0.64,1.30) | 0.60 | | | | 0.93(0.69,1.24) | 0.61 | | | |
| Upper Index | 0.21(0.93, 1.58) | 0.16 | | | | 0.98(0.79,1.22) | 0.88 | | | |
| **Ethnicity** | | | | | | | | | | |
| Advantaged Janajatis | Ref | | Ref | | | Ref | | Ref | | |
| Dalits | 0.53(0.31, 0.89) | **0.02** | 0.71(0.44, 1.14) | 0.15 | 1.0 | 0.35(0.15, 0.79) | **0.01** | 0.49(0.20, 1.17) | 0.11 | 1.0 |
| Disadvantaged Janajatis | 0.44(0.29, 0.68) | **<0.001** | 0.52(0.36, 0.78) | **0.001** | **0.029** | 0.43(0.22, 0.85) | **0.02** | 0.45(0.22, 0.92) | **0.028** | 0.539 |
| Religious Minorities | 0.7(0.37, 1.34) | 0.28 | 1.07(0.54, 2.14) | 0.84 | 1.0 | 0.47(0.10, 2.18) | 0.33 | 0.85(0.18, 4.04) | 0.84 | 1.0 |

*(Continued)*

**Table 3.** (Continued)

| Characteristics | Overweight Vs Normal | | | | | Obesity Vs Normal | | | | |
| | Unadjusted | | Adjusted | | Bonferroni p-value | Unadjusted | | Adjusted | | Bonferroni p-value |
| | OR (95% CI) | p-value | OR (95% CI) | p-value | | OR (95% CI) | p-value | OR (95% CI) | p-value | |
| Upper caste | 0.66(0.42, 1.05) | 0.08 | 0.85(0.57, 1.27) | 0.43 | 1.0 | 0.57(0.32, 1.03) | 0.06 | 0.75(0.40, 1.40) | 0.37 | 1.0 |
| **Marital Status** | | | | | | | | | | |
| Unmarried | Ref | | Ref | | | Ref | | Ref | | |
| Ever Married | 3.07(1.73, 5.44) | **<0.001** | 2.11(1.07, 4.13) | **0.03** | 0.63 | 3.41(1.03, 11.3) | **0.05** | 1(0.22, 4.50) | 0.99 | 1.0 |
| **Occupation** | | | | | | | | | | |
| Unemployed | Ref | | Ref | | | Ref | | Ref | | |
| Students | 0.32(0.11, 0.96) | **0.04** | 0.78(0.31, 1.97) | 0.59 | 1.0 | 0.1(0.01, 0.66) | **0.02** | 0.18(0.02, 1.56) | 0.12 | 1.0 |
| Home maker | 0.96(0.42, 2.22) | 0.93 | 1.21(0.66, 2.19) | 0.53 | 1.0 | 1.34(0.40, 4.54) | 0.64 | 1.65(0.56, 4.84) | 0.36 | 1.0 |
| Employed | 1.15(0.48, 2.78) | 0.75 | 1.5(0.80, 2.81) | 0.2 | 1.0 | 1.92(0.55, 6.70) | 0.3 | 2.6(0.91, 7.42) | 0.073 | 1.0 |
| Others | 0.67(0.26, 1.73) | 0.41 | 0.9(0.42, 1.90) | 0.77 | 1.0 | 0.52(0.11, 2.45) | 0.41 | 0.91(0.22, 3.79) | 0.9 | 1.0 |
| **Current Smoking** | | | | | | | | | | |
| No | Ref | | | | | Ref | | | | |
| Yes | 0.9(0.61, 1.33) | 0.6 | | | | 0.59(0.33, 1.07) | 0.08 | | | |
| **Alcohol Consumption** | | | | | | | | | | |
| No | Ref | | | | | Ref | | | | |
| Yes | 1.12(0.83, 1.50) | 0.45 | | | | 0.78(0.48, 1.28) | 0.33 | | | |
| **Sufficient Fruits and Vegetables** | | | | | | | | | | |
| Insufficient | Ref | | | | | Ref | | | | |
| Sufficient | 1.19(0.76, 1.84) | 0.44 | | | | 0.87(0.43, 1.75) | 0.69 | | | |
| **Sufficient Physical Activity** | | | | | | | | | | |
| Insufficient | Ref | | | | | Ref | | Ref | | |
| Sufficient | 0.74(0.53, 1.03) | 0.08 | | | | 0.57(0.36, 0.91) | **0.02** | 0.63(0.39, 1.02) | 0.061 | 1.0 |
| **24 hours Salt Intake** | | | | | | | | | | |
| No | | | | | | | | | | |
| Yes | 1.5(1.03, 2.18) | **0.03** | 1.39(0.86, 2.24) | 0.17 | 1.0 | 0.96(0.58, 1.59) | 0.88 | | | |
| **Cholesterol** | 1.01(1.00, 1.01) | **<0.001** | 1(1.00, 1.01) | **0.01** | 0.21 | 1.01(1.01, 1.02) | **<0.001** | 1.01(1.01, 1.02) | **<0.001** | **0.0004** |

home makers (AOR = 2.15, 95% CI = 1.12-4.11, p = 0.021) and the "other" category (AOR = 3.14, 95% CI = 1.31-7.52, p = 0.011) had the higher odds of prehypertension as compared to the unemployed. However, after applying Bonferroni correction, both associations were no longer statistically significant. Participants with high cholesterol had higher odds of being pre-hypertension (AOR = 1.00, 95% CI = 1.00–1.02, p =0.001) and higher odds of being hypertension (AOR = 1.03, 95% CI = 1.02-1.04, p = 0.001), this result is significant before and after Bonferroni correction. As compared to normal weight individuals, underweight (AOR = 0.54, 95% CI = 0.34-0.87, p = 0.011) was significantly associated with pre-hypertension (AOR = 1.69, 95% CI = 1.19-2.39, p = 0.003), however, after applying Bonferroni correction, association

**Table 4. Factors associated with Pre-Hypertension and Hypertension in univariate and multivariate logistic regression among Urban Population.**

| Characteristics | Pre-Hypertension Vs Normal | | | | | Hypertension Vs Normal | | | | |
|---|---|---|---|---|---|---|---|---|---|---|
| | Unadjusted | | Adjusted | | Bonferroni p-value | Unadjusted | | Adjusted | | Bonferroni p-value |
| | OR (95% CI) | p-value | OR (95% CI) | p-value | | OR 95% CI | p-value | OR 95% CI | p-value | |
| **Age*** | 1.03(1.02, 1.04) | **<0.001** | 1.01(1.00, 1.02) | 0.11 | 1.0 | 1.05(1.04, 1.06) | **<0.001** | 1.03(1.02, 1.04) | **<0.001** | **<0.001** |
| **Sex** | | | | | | | | | | |
| Women | Ref | | | | | Ref | | | | |
| Men | 1.41(1.09, 1.84) | **0.01** | 1.68(1.19, 2.36) | **0.003** | **0.05** | 1.98(1.53, 2.57) | **<0.001** | 2.33(1.56, 3.47) | **<0.001** | **<0.001** |
| **Ecological Region** | | | | | | | | | | |
| Mountain | Ref | | | | | Ref | | | | |
| Hilly | 1.33(0.70, 2.52) | 0.38 | | | | 1.33(0.71, 2.46) | 0.37 | | | |
| Tarai | 1.37(0.73, 2.58) | 0.32 | | | | 1.04(0.58, 1.85) | 0.91 | | | |
| **Province** | | | | | | | | | | |
| Koshi | Ref | | | | | Ref | | | | |
| Madhesh | 1.1(0.66, 1.85) | 0.71 | | | | 0.82(0.42, 1.57) | 0.54 | | | |
| Bagmati | 0.7(0.42, 1.17) | 0.17 | | | | 0.93(0.48, 1.79) | 0.83 | | | |
| Gandaki | 1.16(0.75, 1.81) | 0.5 | | | | 1.34(0.74, 2.40) | 0.33 | | | |
| Lumbini | 1.08(0.64, 1.81) | 0.77 | | | | 1.25(0.68, 2.32) | 0.47 | | | |
| Karnali | 1.1(0.68, 1.77) | 0.69 | | | | 0.84(0.43, 1.63) | 0.6 | | | |
| Sudurpashchim | 1.11(0.70, 1.74) | 0.66 | | | | 0.88(0.45, 1.72) | 0.71 | | | |
| **Education** | | | | | | | | | | |
| No formal education | Ref | | | | | Ref | | Ref | | |
| Primary | 1.14(0.79, 1.65) | 0.47 | | | | 0.84(0.63, 1.13) | 0.24 | 1.14(0.80, 1.62) | 0.48 | 1.0 |
| Secondary | 0.84(0.61, 1.16) | 0.28 | | | | 0.39(0.28, 0.54) | **<0.001** | 0.65(0.41, 1.03) | 0.065 | 1.0 |
| University | 0.52(0.24, 1.13) | 0.1 | | | | 0.51(0.22, 1.20) | 0.12 | 0.71(0.22, 2.26) | 0.56 | 1.0 |
| **Wealth Quintile** | | | | | | | | | | |
| Lower Index | Ref | | | | | Ref | | | | |
| Middle Index | 1.14(0.84,1.56) | 0.39 | | | | 0.97(0.67,1.41) | 0.88 | | | |
| Upper Index | 1.16(0.91,1.47) | 0.24 | | | | 0.99(0.75,1.29) | 0.92 | | | |
| **Ethnicity** | | | | | | | | | | |
| Advantaged Janajatis | Ref | | Ref | | | Ref | | | | |
| Dalits | 1.25(0.67, 2.33) | 0.49 | 1.23(0.67, 2.26) | 0.51 | 1.0 | 1.45(0.88, 2.39) | 0.15 | | | |
| Disadvantaged Janajatis | 1.2(0.75, 1.94) | 0.44 | 1.22(0.73, 2.01) | 0.45 | 1.0 | 0.75(0.47, 1.20) | 0.23 | | | |
| Religious Minorities | 1.65(0.96, 2.82) | 0.07 | 1.51(0.81, 2.83) | 0.19 | 1.0 | 0.89(0.48, 1.67) | 0.72 | | | |
| Upper caste | 1.56(1.00, 2.45) | 0.05 | 1.78(1.13, 2.79) | **0.013** | 0.201 | 0.88(0.57, 1.37) | 0.57 | | | |
| **Marital Status** | | | | | | | | | | |
| Unmarried | Ref | | Ref | | | Ref | | Ref | | |

*(Continued)*

**Table 4.** (Continued)

| Characteristics | Pre-Hypertension Vs Normal | | | | | Hypertension Vs Normal | | | | |
|---|---|---|---|---|---|---|---|---|---|---|
| | Unadjusted | | Adjusted | | Bonferroni p-value | Unadjusted | | Adjusted | | Bonferroni p-value |
| | OR (95% CI) | p-value | OR (95% CI) | p-value | | OR 95% CI | p-value | OR 95% CI | p-value | |
| Ever Married | 1.75(1.21, 2.54) | **<0.001** | 1.27(0.63, 2.54) | 0.5 | 1.0 | 3.76(2.29, 6.18) | **<0.001** | 0.73(0.39, 1.40) | 0.34 | 1.0 |
| **Occupation** | | | | | | | | | | |
| Unemployed | Ref | | Ref | | | Ref | | Ref | | |
| Students | 1.06(0.51, 2.19) | 0.87 | 1.61(0.61, 4.21) | 0.33 | 1.0 | 0.1(0.04, 0.25) | **<0.001** | 0.25(0.09, 0.68) | **0.007** | 0.119 |
| Home maker | 1.81(1.00, 3.25) | **0.05** | 2.15(1.12, 4.11) | **0.021** | 0.34 | 1.32(0.73, 2.38) | 0.35 | 1.62(0.90, 2.90) | 0.1 | 1.0 |
| Employed | 1.86(0.99, 3.46) | **0.05** | 1.68(0.85, 3.32) | 0.14 | 1.0 | 2.3(1.28, 4.12) | **0.01** | 1.82(0.96, 3.44) | 0.066 | 1.0 |
| Others | 2.9(1.33, 6.32) | **0.01** | 3.14(1.31, 7.52) | **0.011** | 0.17 | 1.69(0.79, 3.60) | 0.18 | 2.07(0.92, 4.66) | 0.079 | 1.0 |
| **Current Smoking** | | | | | | | | | | |
| No | Ref | | | | | Ref | | Ref | | |
| Yes | 0.59(0.33, 1.07) | 0.08 | | | | 1.76(1.28, 2.42) | **<0.001** | 0.86(0.58, 1.27) | 0.45 | 1.0 |
| **Alcohol Consumption** | | | | | | | | | | |
| No | Ref | | | | | Ref | | Ref | | |
| Yes | 1.29(0.93, 1.79) | 0.13 | | | | 2.34(1.82, 3.02) | **<0.001** | 1.58(1.08, 2.31) | **0.019** | 0.321 |
| **Sufficient Fruits and Vegetables** | | | | | | | | | | |
| Insufficient | Ref | | | | | Ref | | | | |
| Sufficient | 1.35(0.92, 1.97) | 0.12 | | | | 1.06(0.73, 1.54) | 0.76 | | | |
| **Sufficient Physical Activity** | | | | | | | | | | |
| Insufficient | Ref | | | | | Ref | | | | |
| Sufficient | 1.06(0.77, 1.46) | 0.71 | | | | 1.31(0.93, 1.86) | 0.13 | | | |
| **24 hours Salt Intake** | | | | | | | | | | |
| No | Ref | | | | | Ref | | | | |
| Yes | 0.98(0.67, 1.43) | 0.92 | | | | 0.91(0.65, 1.28) | 0.58 | | | |
| **Cholesterol** | 1.03(1.02, 1.04) | **<0.001** | 1.01(1.00, 1.02) | **0.001** | **0.019** | 1.05(1.04, 1.06) | **<0.001** | 1.03(1.02, 1.04) | **<0.001** | **<0.001** |
| **BMI categories** | | | | | | | | | | |
| Normal | Ref | | Ref | | | Ref | | Ref | | |
| Obesity | 2.15(1.27, 3.64) | **<0.001** | 1.65(0.94, 2.90) | 0.083 | 1.0 | 5.85(3.71, 9.24) | **<0.001** | 5.12(2.95, 8.87) | **<0.001** | **<0.001** |
| Overweight | 1.23(0.85, 1.79) | 0.27 | 1.01(0.69, 1.48) | 0.95 | 1.0 | 1.88(1.40, 2.51) | **<0.001** | 1.69(1.19, 2.39) | **0.003** | 0.0553 |
| Underweight | 0.55(0.35, 0.86) | **0.01** | 0.54(0.34, 0.87) | **0.011** | 0.183 | 0.61(0.37, 1.01) | 0.06 | 0.6(0.36, 1.00) | 0.05 | 0.848 |

was no longer statistically significant. Being obesity (AOR = 5.12, 95% CI = 2.95-8.87, p = 0.001) and obese (AOR = 1.69, 95% CI = 1.19-2.39, p = 0.003) were significantly associated with hypertension. However, these results remain statistically significant for obesity after Bonferroni correction (Bonferroni p<0.001).

Table 5 shows the factors associated with pre-diabetics and diabetics among urban population. Ecological region and obesity showed significant associations with pre-diabetes. Age, ecological region, province, education, ethnicity, cholesterol and obesity were found significantly associated with diabetes. For each one-year increase in age, the odds of having diabetes increased by approximately 2% (AOR = 1.02, 95% CI = 1.01-1.04, p = 0.011), however, after Bonferroni correction (Bonferroni p = 0.553), this result is not statistically significant. Compared to the mountain region, urban population residing in the hilly region had substantially higher odds of pre-diabetes (AOR = 3.59, 95% CI = 1.6-8.08, p = 0.002) and diabetes (AOR = 6.44, 95% CI = 3.31-11.10, p = 0.002) those in the Tarai region had higher odds (AOR = 5.07, 95% CI = 2.44-10.5, p < 0.001) of pre-diabetes and diabetics (AOR = 5.96, 95% CI = 3.12-19.86, p = 0.001). however, after Bonferroni correction, the results were found to significant for Tarai region. Urban population living in Madhesh province had higher odds of being diabetic (AOR = 3.1, 95% CI = 1.23-7.83, p = 0.017) compared to the Koshi province but after Bonferroni correction (Bonferroni p = 0.23), this result was not statistically significant. University level education had higher odds of being diabetic (AOR = 3.98, 95% CI = 1.39-11.44, p = 0.011 compared to no formal education and religious minorities had the highest odds of diabetes (AOR = 3.76, 95% CI = 1.11-12.73, p = 0.034) compared to advantaged Janajatis, but the results was not significant after Bonferroni correction. However, after applying Bonferroni correction, both associations were no longer statistically significant. Participants with high cholesterol had higher odds of being pre-diabetics (AOR = 1.00, 95% CI = 1.00–1.02, p <0.001) and higher odds of being Pre-diabetics (AOR = 1.03, 95% CI = 1.02-1.04, p <0.001), this result is significant before and after Bonferroni correction. Being obese was also significantly associated with being pre-diabetic (AOR=1.82, 95% CI=1.05-3.16, P=0.033) and diabetes AOR=2.41, 95% CI= 1.06-5.48, p=0.037), however, the results were not statistically significant after Bonferroni correction.

## Discussion

Overall, our analysis highlights the association between socio-economic determinants, NCD risk behaviors and NCD within the urban population. Our analysis revealed notable disparities in education and occupation within the urban population, showing the socio-economic factors contributing to these differences. As revealed in the analysis, those with higher levels of education and employment are more prevalent in the middle and rich quintiles, whereas those with no formal education and homemakers are more prevalent in the poor quintile. Most urban population live in slums that are unregulated, have congested conditions, are overcrowded, are positioned near open sewers, and restricted to geographically dangerous areas such as hillsides, riverbanks, and water basins subject to flooding [10,13]. This has contributed to a growing gap between rich and poor in terms of adequate urban housing, employment opportunities, transportation, levels of education, and access to affordable health services of decent quality [12,13,19].

Our study also identified several social determinants and risk factors associated with non-communicable diseases among the urban population in Nepal. To ensure the robustness of our findings, we applied Bonferroni correction to adjust for the multiple comparisons in our analysis. This correction ensures that our reported associations are not due to chance. While marital status and overweight, education and diabetes, as well as homemakers, employment, and prehypertension initially showed associations in the adjusted logistic

regression, these relationships were not statistically significant after applying the Bonferroni correction. This indicates that the observed associations may have been influenced by multiple comparisons and should be interpreted with caution.

Age emerged as a significant predictor for hypertension after applying the Bonferroni correction, and these findings aligns with well documents public health trends. As individuals age, several physiological changes, such as arterial stiffening, increased vascular resistance, and decreased kidney function, contribute to higher blood pressure levels [20,21]. Arteries lose their elasticity, which leads to increased resistance and forces the heart to work harder, raising blood pressure. Additionally, age-related decline in kidney function impairs the body's ability to regulate sodium and fluid balance, further increasing the risk of hypertension [20–22]. From a public health perspective, aging is also associated with the accumulation of lifestyle-related risk factors, including reduced physical activity, poor dietary habits, and increased obesity, all of which contribute to hypertension [21,22].

**Table 5. Factors associated with Pre-Diabetics and Diabetics in univariate and multivariate logistic regression among Urban Population.**

| Characteristics | Pre-Diabetics Vs Normal | | | | | Diabetics Vs Normal | | | | |
|---|---|---|---|---|---|---|---|---|---|---|
| | Unadjusted | | Adjusted | | Bonferroni p-value | Unadjusted | | Adjusted | | Bonferroni p-value |
| | OR 95% CI | p-value | OR 95% CI | p-value | | OR 95% CI | p-value | OR 95% CI | p-value | |
| **Age*** | 1.02(1.01, 1.03) | **<0.001** | 1.01(1.00, 1.02) | 0.11 | 1.0 | 1.04(1.02, 1.05) | **<0.001** | 1.02(1.01, 1.04) | **0.011** | 0.533 |
| **Sex** | | | | | | | | | | |
| Women | Ref | | | | | Ref | | | | |
| Men | 1.15(0.88, 1.51) | 0.29 | | | | 1.12(0.75, 1.68) | 0.58 | | | |
| **Ecological Region** | | | | | | | | | | |
| Mountain | Ref | | Ref | | | Ref | | Ref | | |
| Hilly | 2.99(1.21, 7.39) | **0.02** | 3.59(1.6, 8.08) | **0.002** | **0.032** | 3.88(0.66, 22.9) | 0.13 | 6.44(3.31, 11.10) | **0.002** | 0.074 |
| Tarai | 4.37(1.90, 10.1) | **<0.001** | 5.07(2.44, 10.5) | **<0.001** | **<0.001** | 5.84(1.43, 8.96) | **<0.001** | 5.96(3.12, 19.86) | **<0.001** | **0.002** |
| **Province** | | | | | | | | | | |
| Koshi | | | | | | | | | | |
| Madhesh | 1.1(0.66, 1.85) | 0.71 | | | | 3.52(1.51, 8.20) | **<0.001** | 3.1(1.23, 7.83) | **0.017** | 0.23 |
| Bagmati | 0.7(0.42, 1.17) | 0.17 | | | | 0.83(0.34, 2.01) | 0.67 | 1.15(0.51, 2.56) | 0.74 | 1.0 |
| Gandaki | 1.16(0.75, 1.81) | 0.5 | | | | 0.78(0.30, 1.99) | 0.59 | 1.03(0.43, 2.49) | 0.94 | 1.0 |
| Lumbini | 1.08(0.64, 1.81) | 0.77 | | | | 1.85(0.77, 4.45) | 0.17 | 1.26(0.58, 2.70) | 0.56 | 1.0 |
| Karnali | 1.1(0.68, 1.77) | 0.69 | | | | 0.1(0.03, 0.29) | **<0.001** | 0.28(0.08, 1.01) | 0.052 | 1.0 |
| Sudurpash-chim | 1.11(0.70, 1.74) | 0.66 | | | | 1(0.28, 3.64) | 1 | 1.15(0.23, 5.70) | 0.86 | 1.0 |
| **Education** | | | | | | | | | | |
| No formal education | Ref | | Ref | | | Ref | | Ref | | |
| Primary | 1.02(0.71, 1.49) | 0.9 | 1.24(0.80, 1.92) | 0.34 | 1.0 | 0.79(0.46, 1.33) | 0.36 | 1.47(0.76, 2.84) | 0.25 | 1.0 |
| Secondary | 0.64(0.45, 0.93) | **0.02** | 0.84(0.55, 1.27) | 0.4 | 1.0 | 0.49(0.30, 0.81) | **0.01** | 1.17(0.61, 2.23) | 0.64 | 1.0 |
| University | 0.43(0.25, 0.75) | **<0.001** | 0.61(0.33, 1.13) | 0.11 | 1.0 | 0.91(0.35, 2.37) | 0.85 | 3.98(1.39, 11.44) | **0.011** | **0.763** |
| **Wealth Quintile** | | | | | | | | | | |
| Lower Index | Ref | | | | | Ref | | | | |
| Middle Index | 1.28(0.92,1.78) | 0.14 | | | | 0.97(0.67,1.41) | 0.88 | | | |
| Upper Index | 1.26(0.95,1.66) | 0.10 | | | | 0.99(0.75,1.29) | 0.92 | | | |
| **Ethnicity** | | | | | | | | | | |
| Advantaged Janajatis | Ref | | | | | Ref | | Ref | | |
| Dalits | 0.91(0.45, 1.82) | 0.78 | | | | 1.16(0.43, 3.14) | 0.76 | 1.18(0.42, 3.28) | 0.75 | 1.0 |
| Disadvan-taged categories | 1.13(0.61, 2.08) | 0.7 | | | | 2.15(1.14, 4.08) | **0.02** | 1.44(0.75, 2.77) | 0.28 | 1.0 |
| Religious Minorities | 1.17(0.52, 2.63) | 0.7 | | | | 6.22(2.39, 16.1) | **<0.001** | 3.76(1.11, 12.73) | **0.034** | 1.0 |
| Upper caste | 0.8(0.44, 1.47) | 0.47 | | | | 1.32(0.52, 3.35) | 0.56 | 1.18(0.54, 2.57) | 0.68 | 1.0 |

*(Continued)*

| Characteristics | Pre-Diabetics Vs Normal | | | | | Diabetics Vs Normal | | | | |
| --- | --- | --- | --- | --- | --- | --- | --- | --- | --- | --- |
| | Unadjusted | | Adjusted | | Bonferroni p-value | Unadjusted | | Adjusted | | Bonferroni p-value |
| | OR 95% CI | p-value | OR 95% CI | p-value | | OR 95% CI | p-value | OR 95% CI | p-value | |
| **Marital Status** | | | | | | | | | | |
| Unmarried | Ref | | Ref | | | Ref | | Ref | | |
| Ever Married | 1.62(1.11, 2.34) | **0.01** | 1.0(0.66, 1.52) | 0.98 | 1.0 | 3.8(1.54, 9.38) | **<0.001** | 1.31(0.23, 7.54) | 0.76 | |
| **Occupation** | | | | | | | | | | |
| Unemployed | Ref | | | | | Ref | | Ref | Ref | |
| Students | 0.65(0.27, 1.56) | 0.33 | | | | 0.52(0.14, 1.92) | 0.33 | 0.86(0.10, 7.13) | 0.89 | 1.0 |
| Home maker | 1.46(0.71, 3.02) | 0.31 | | | | 2.11(0.73, 6.09) | 0.17 | 1.23(0.41, 3.67) | 0.71 | 1.0 |
| Employed | 1.7(0.82, 3.52) | 0.15 | | | | 1.38(0.48, 3.97) | 0.54 | 0.81(0.28, 2.34) | 0.7 | 1.0 |
| Others | 2.01(0.93, 4.34) | 0.07 | | | | 3.1(1.03, 9.31) | 0.04 | 1.76(0.55, 5.67) | 0.34 | 1.0 |
| **Current Smoking** | | | | | | | | | | |
| No | Ref | | | | | Ref | | | | |
| Yes | 1.19(0.86, 1.65) | 0.3 | | | | 1.54(0.86, 2.75) | 0.15 | | | |
| **Alcohol Consumption** | | | | | | | | | | |
| No | Ref | | | | | Ref | | | | |
| Yes | 1.03(0.77, 1.38) | 0.85 | | | | 0.92(0.55, 1.54) | 0.75 | | | |
| **Sufficient Fruits and Vegetables** | | | | | | | | | | |
| Insufficient | Ref | | | | | Ref | | | | |
| Sufficient | 1.44(0.99, 2.09) | 0.05 | 1.3(0.88, 1.92) | 0.19 | 1.0 | 1.53(0.76, 3.05) | 0.23 | | | |
| **Sufficient Physical Activity** | | | | | | | | | | |
| Insufficient | Ref | | | | | Ref | | | | |
| Sufficient | 0.85(0.60, 1.23) | 0.39 | | | | 0.98(0.59, 1.63) | 0.94 | | | |
| **24 hours Salt Intake** | | | | | | | | | | |
| No | Ref | | | | | Ref | | | | |
| Yes | 1.39(0.92, 2.10) | 0.11 | | | | 0.88(0.51, 1.54) | 0.66 | | | |
| **Cholesterol** | 1.01(1.01, 1.02) | **<0.001** | 1.01(1.01, 1.01) | **<0.001** | **<0.001** | 1.02(1.01, 1.02) | **<0.001** | 1.01(1.01, 1.02) | **<0.001** | **<0.001** |
| **Obesity** | | | | | | | | | | |
| Normal | Ref | | Ref | | | Ref | | Ref | | |
| Obesity | 2.21(1.32, 3.68) | **<0.001** | 1.82(1.05, 3.16) | **0.033** | 0.497 | 3.3(1.63, 6.70_ | **<0.001** | 2.41(1.06, 5.48) | **0.027** | **0.631** |
| Overweight | 1.22(0.88, 1.68) | 0.23 | 1.1(0.81, 1.49) | 0.53 | 1.0 | 1.32(0.68, 2.58) | 0.41 | 1.06(0.46, 2.45) | 0.9 | 1.0 |
| Underweight | 1.01(0.67, 1.53) | 0.95 | 0.96(0.64, 1.45) | 0.86 | 1.0 | 0.57(0.29, 1.12) | 0.1 | 0.59(0.28, 1.24) | 0.16 | 1.0 |
| **Hypertension** | | | | | | | | | | |
| Normal | Ref | | Ref | | | Ref | | Ref | | |
| Pre-Hypertension | 1.32(0.95, 1.84) | 0.1 | 1.12(0.79, 1.58) | 0.53 | 1.0 | 1.98(1.01, 3.90) | **0.05** | 1.44(0.74, 2.84) | 0.28 | 1.0 |
| Hypertension | 1.49(1.08, 2.06) | **0.02** | 1.06(0.74, 1.52) | 0.74 | 1.0 | 2.93(1.59, 5.38) | **<0.001** | 1.84(0.99, 3.41) | 0.054 | 1.0 |

Sex-based differences were evident, with men exhibiting higher odds of pre-hypertension and hypertension compared to women. Epidemiological data consistently demonstrate a higher prevalence of hypertension in males compared to females across various age groups [22,23]. This discrepancy may be attributed to hormonal differences, as well as lifestyle and behavioral factors more prevalent in males, such as higher rates of tobacco and alcohol consumption as shown by studies [14,23,24]. Additionally, genetic predispositions and variations in physiological responses to stressors may contribute to this gender disparity [23,24]. However, contradictory findings were observed in a recent study, where female participants from India were more likely to be hypertensive than males, in comparison to women from Bangladesh and Nepal [24].

The association between ethnicity and the risk of obesity underscores the role of social determinants in health disparities. Disadvantaged ethnic groups in Nepal were found to have significantly lower odds of being obesity compared to advantaged groups, suggesting that socio-economic status, access to healthcare, and health literacy might be influencing these outcomes [14,25]. These findings align with previous research that highlights how marginalized communities often face both protective and risk factors differently due to their socio-economic conditions and limited access to resources [3,14,25].

Likewise, urban populations residing in Tarai region had substantially higher odds of pre-diabetes and diabetes as compared to those residing in mountain region can be attributed to a combination of lifestyle, environmental, and socio-economic factors unique to urban settings. According to recent reports, 53.66% of the population resides in the Tarai region, while 40.25% live in hilly areas, and 6.09% inhabit mountainous regions [16]. This demographic distribution is crucial, as the density of the population in the Tarai contributes to the challenges associated with urbanization [16,26]. Urban areas often exhibit lifestyle changes that contribute to the prevalence of diabetes, including increased consumption of processed and high-calorie foods, lower levels of physical activity, and higher rates of obesity [12,15,27]. In the Tarai region, rapid urbanization has led to changes in dietary patterns, with greater access to unhealthy food options and reduced physical activity due to sedentary lifestyles. This shift is often exacerbated by stressors associated with urban living, such as economic pressures and increased competition for resources [15,26,28]. In contrast, the Tarai region's flatter topography and greater accessibility to modern amenities may lead to a higher prevalence of sedentary lifestyles and less healthy dietary practices, thereby elevating diabetes prevalence in that region [12,27,28].

Additionally, the urban environment may facilitate higher exposure to risk factors such as air pollution and limited access to green spaces, which can also negatively impact metabolic health [11,13]. In contrast, mountain populations may engage in more physically demanding lifestyles, such as walking long distances and farming, which can contribute to lower rates of obesity and better overall metabolic health [27,29].

Furthermore, health infrastructure and access to healthcare services can differ significantly between urban and rural areas [30,31]. Urban residents in the Tarai may have better access to diagnostic and treatment facilities, potentially leading to higher rates of diagnosis for pre-diabetes and diabetes [30,31]. However, this access can also mean that urban populations are more likely to be screened and diagnosed, which can contribute to observed higher prevalence rates [30.31].

Urban population in Karnali and Sudurpashchim provinces had lower odds of overweight compared to those in Koshi province can be attributed to a range of lifestyle, dietary, and socio-economic factors that characterize these regions [14,15,32]. Firstly, the dietary habits of urban populations in Karnali and Sudurpashchim provinces may differ significantly from those in Koshi. These regions often have more traditional dietary patterns, with a higher reliance on locally sourced, whole foods, such as grains, vegetables, and legumes [33]. In contrast, urban areas in Koshi may have greater access to processed and high-calorie foods, which can contribute to higher rates of overweight and obesity. The availability of fast food and convenience foods in urban settings is often linked to dietary changes that promote weight gain [33,34]. Moreover, socio-economic factors play a crucial role in influencing weight status. The economic conditions in Karnali and Sudurpashchim provinces may lead to less disposable income for purchasing unhealthy foods or engaging in sedentary leisure activities [10,14,33]. Additionally, cultural perceptions around body weight and health may differ, with a greater emphasis on maintaining traditional lifestyles that promote healthier body weights in these regions [33–35].

The association between cholesterol and obesity, hypertension, and diabetes underscores the complex interplay of these risk factors in our study [36,37]. Elevated cholesterol often coexists with these conditions due to shared underlying mechanisms. Obesity can lead to dyslipidemia and impaired lipid metabolism, resulting in elevated cholesterol levels [36,38]. Similarly, hypertension and diabetes can disrupt lipid profiles through metabolic pathways and oxidative stress. Additionally, these conditions can collectively contribute to atherosclerosis, further elevating cholesterol levels [37,39].

Obesity has been consistently linked to hypertension through various mechanisms and studies, highlighting the multifactorial nature of this relationship. The association between obesity and hypertension is well-documented, with numerous studies indicating that increased body mass index (BMI) correlates with elevated blood pressure levels. For instance, a study in Pakistan revealed that 39% of overweight and 19.5% of obese participants had hypertension, underscoring the critical role of obesity as a modifiable risk factor for hypertension [40]. Similarly, research conducted in China demonstrated a significant prevalence of obesity-related hypertension among middle-aged and older adults, particularly noting that obesity was associated with higher incidences of diabetes and dyslipidemia, both of which are risk factors for cardiovascular diseases [41]. Moreover, the relationship between obesity and hypertension is not merely correlative but also indicative of a potential causal pathway. Studies have shown that the risk of developing hypertension increases with higher BMI, with a dose-dependent relationship observed across various populations [37–41]. For instance, a meta-analysis indicated that the prevalence of hypertension rises significantly with increasing obesity levels, reinforcing the notion that obesity is a critical risk factor for hypertension [27,35].

This study has few limitations. Urban areas in Nepal vary significantly in terms of infrastructure, healthcare access, and environmental factors, making it challenging to generalize the findings across all urban populations. Additionally, the study focuses exclusively on urban populations, excluding rural areas, which limits the generalizability of the findings to the entire country. Moreover, the STEPS survey used in the study may not capture all relevant socio-economic variables, such as detailed income data and access to healthcare services, which could restrict a comprehensive analysis of wealth-related disparities.

## Conclusion

In conclusion, our analysis underscores the intricate interplay between socio-economic determinants, NCD risk behaviors, and NCDs within urban populations. Factors such as education, occupation, age, gender, ecological region, and province contribute significantly to the prevalence of conditions like hypertension, diabetes, and overweight among the urban population. The disparities observed in different regions suggest the influence of diverse lifestyles, socio-economic conditions, and access to healthcare services. Understanding these dynamics is crucial for developing targeted interventions to address the complex web of factors contributing to the burden of NCDs among urban populations. Public health efforts should aim to create comprehensive strategies that address socio-economic disparities, promote healthy lifestyles, and enhance health literacy to effectively combat the rising prevalence of NCDs.

## Supporting information

**S1 Table. Wealth Categories in Urban Population**
(DOCX)

## Author contributions

**Conceptualization:** Sampurna Kakchapati, Helen Elsey, Sushil Chandra Baral.

**Data curation:** Raju Neupane.

**Formal analysis:** Sampurna Kakchapati, Kriti Sagar Baral.

**Investigation:** Grishu Shrestha, Deepak Joshi.

**Methodology:** Raju Neupane.

**Project administration:** Grishu Shrestha.

**Supervision:** Raju Neupane, Tim Ensor.

**Validation:** Kriti Sagar Baral, Bryony Dawkins.

**Visualization:** Deepak Joshi, Tim Ensor.

**Writing – original draft:** Tim Ensor, Helen Elsey.

**Writing – review & editing:** Sampurna Kakchapati, Bryony Dawkins, Helen Elsey, Sushil Chandra Baral.

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
