## [Decision Letter · Decision Letter 0]

26 Aug 2024

PONE-D-24-27355Social determinants and risk factors associated with non-communicable diseases among urban population in Nepal: A comparative study of poor, middle and rich wealth categories of urban population using STEPS surveyPLOS ONE

Dear Dr. Kakchapati,

Thank you for submitting your manuscript to PLOS ONE. After careful consideration, we feel that it has merit but does not fully meet PLOS ONE’s publication criteria as it currently stands. Therefore, we invite you to submit a revised version of the manuscript that addresses the points raised during the review process.

Please carefully address all the concerns expressed by reviewers and editor's feedback.*Terai * is fine to keep though Reviewer suggested *Tarai * as both are being used. 

We look forward to receiving your revised manuscript.

Kind regards,

Rajendra Prasad Parajuli, PhD

Academic Editor

PLOS ONE

Additional Editor Comments:

Editors' Feedback:

Version: 1                                                  Date: Aug 25, 2024

MS: PONE-D-24-27355

Title: *Social determinants and risk factors linked to non-communicable diseases among urban populations in Nepal: A comparative analysis across poor, middle, and affluent wealth groups using the STEPS survey* .

General Feedback:

In this study, the authors examined the prevalence and contributing factors of non-communicable diseases (NCDs) within various wealth groups in Nepal.

This research introduces new data regarding the prevalence and relationship between NCDs and contextual factors, highlighting a crucial area of exploration. Nevertheless, the authors should address concerns regarding potential chance findings due to the extensive number of associations analyzed. To mitigate this, authors may consider applying a Bonferroni correction.

Despite these issues, the study offers valuable data and insights. I therefore suggest resubmitting the paper with revisions to address these points.

**Typographical Errors**

**Abstract:**

• Urban population Hilly region → Urban population from the Hilly region

**Introduction:**

**• Typographical errors:**

o Socio-economic factors are associated with behavioral risk factors ..Remove "for" and mortality from NCDs.

• The phrases "Rapid urbanization, connection with vulnerable populations, and elevated NCDs risk" are unclear and challenging for readers. Please clarify what the authors intend to convey, particularly why the detection rate is increasing.

• The statement "Urban residents face material constraints, limiting access to nutritious food, quality healthcare, and the ability to maintain healthy lifestyles" requires further elaboration. How do these constraints manifest?

• The authors should expand their literature review to encompass the latest epidemiological knowledge, particularly in relation to the social determinants of NCDs, both in Nepal and internationally. This will help identify gaps in knowledge and strengthen the rationale for the study, as reviewers have also suggested.

**Methodology:**

• In the Site selection section, numerous data points lack citations.

• Clarify the term "weighted" in the context of multivariate logistic regression models.

• Include the NHRC Ethical Approval Number.

**Results:**

• Given the extensive number of associations examined, there is a high likelihood of chance findings. Therefore, applying Bonferroni correction would be advisable.

**Discussion:**

• The interpretation that "*This finding underscores the potential vulnerability of all urban segments to NCDs and emphasizes the need for targeted interventions and policies across diverse urban populations* " does not align with the results and previous statements. Please revise.

• The statement "*This discrepancy may be attributed to hormonal differences, as well as lifestyle and behavioral factors more prevalent in males, such as higher rates of tobacco and alcohol consumption* " should be directly linked to the study data on alcohol and smoking, rather than relying on citations.

• Consider the application of Bonferroni correction, as only a few associations will likely remain significant, which should be the focus of the discussion.

Despite these concerns, the comprehensive analysis provides valuable new insights. I recommend resubmission with revisions to address these points including that of 3 expert reviewers.

Reviewers' comments:

Reviewer's Responses to Questions

**Comments to the Author**

1. Is the manuscript technically sound, and do the data support the conclusions?

Reviewer #1: Yes

Reviewer #2: Yes

Reviewer #3: Partly

2. Has the statistical analysis been performed appropriately and rigorously? 

Reviewer #1: Yes

Reviewer #2: Yes

Reviewer #3: No

3. Have the authors made all data underlying the findings in their manuscript fully available?

Reviewer #1: Yes

Reviewer #2: Yes

Reviewer #3: Yes

4. Is the manuscript presented in an intelligible fashion and written in standard English?

Reviewer #1: Yes

Reviewer #2: Yes

Reviewer #3: Yes

5. Review Comments to the Author

Reviewer #1: Minor revisions to grammar suggested- occasional use of incorrect articles and verbs (a/and/the - was/were/is/are/had/have), extra spaces, incorrect plural versions of words, inconsistencies in capitalization and number formatting (i.e. 1,000 vs 1000), and comma-separated lists (line 202-203). Ensure that "sex" and "gender" are used consistently rather than interchangeably. Line numbering is lost after page 20.

Figure 1 is labeled as Figure 2. Table 1 could be made easier to understand with some formatting revisions. Ensure consistency in the use of the "STEP" acronym- in line 31, the approach is written as "STEP wise", while later in the paper, it becomes clear that a stepwise approach was used (see lines 115 & 123). Check the R Studio version reported in line 184. The definition of the salt consumption variable is not clear (unclear what "for first time" is referring to).

The discussion section could benefit from expanded explanations of contradictory findings. In some areas of the methods and results section, it is not fully clear whether the authors have controlled for age in the logistic regression models (for example, within the hypertension outcome between students and homemakers). Clarification of the models and adjustments to models for each of the two main outcomes would increase understanding of results.

Ensure correct links in reference list- i.e. reference #1 links to a Google Search, rather than the article.

Overall assessment: Minor Revisions

Reviewer #2: Dear Editor and Valued authors

The manuscript “PONE-D-24-27355” entitled “Social determinants and risk factors associated with non-communicable diseases among urban population in Nepal: A comparative study of poor, middle and rich wealth categories of urban population using STEP survey” in itself is well written and represents a comprehensive study in relation to the risk factors for non-communicable diseases (NCDs). While scanning the manuscript thoroughly, it is understood that authors have clearly mentioned the methodological design with multistage stratified technique. One of the crucial parts of this paper is noticed that it has incorporated almost all socio-demographical parameters which might contribute as risk factors for NCDs. However, some suggestions are provided below which would enable further strength of the manuscript and imply its importance in public health domain. This paper can be considered provided the below mentioned aspects are satisfactorily addressed. In a nutshell, a minor revision is required.

Suggestions/Comments:

1. This manuscript is silent regarding the hypothesis of the study which is deemed pertinent in such kind of public health related analytical study. It would be fine to mention the hypothesis briefly in the last paragraph of the “Introduction” section. Please consider it in the revised MS.

2. As mentioned in Table 1, what does the term “average day” indicate? Please specify.

3. The variable “24 hours salt consumption” seems to be poorly defined (Table 1). Please redefine it.

4. The term “Terai” used in the MS should be mentioned as “Tarai” as it is a typical Nepali naming and should look like an original one.

5. In Methods section or elsewhere, a country map of Nepal indicating the proportion of study participants representing the provincial location and ecological belt would be worth incorporating. If possible, the dominant associated risk factors for NCDs be mentioned with specific notation (i.e., with different color code or same color with decreasing intensity) in the specific ecological and political zone of the country. This would provide readers an overall picture of the entire study.

6. Minor typo errors need to be corrected throughout the MS.

Regards,

Reviewer #3: Comments to the authors:

Thank you for the opportunity to review this paper. This manuscript specifically compared the social determinants, NCD risks and prevalence among different wealth categories and determined the factors associated with different NCDs among urban population in Nepal. Given the limited epidemiological evidence on the risk factors and determinants of NCDs in low- and middle-income countries, including Nepal, this study had tried to address a growing concern of the impact of increasing urbanization and how socioeconomic status affects NCD risk and prevalence within urban settings using the recent data from a 2019 STEPS survey from Nepal. However, I have several comments and suggestions which might be helpful to improve the manuscript.

Abstract:

Methods:

• Study design is not mentioned

• You can specify the age group of the participants.

Results: You may include some important result data from table 1

Conclusion: Recommendations are quite general and are not specific based on the study findings. I would suggest making them more specific that align with the study findings.

Minor points:

• In line 32, you can add acronym for STEPwise approach to NCD risk factor surveillance

• In line 41, spell out AOR when it appeared for the first time in the abstract also.

Introduction:

• Please cite more previous studies from Nepal to explain why this study is important among urban population in Nepal.

• In the last paragraph, the authors mentioned “this study aims to inform effective strategies that mitigate the burden of NCDs and enhance the overall well-being of the urban population in Nepal.” Was this also the study objective and how was this done from this study?

• Line 74: grammar, ‘for’ after behavioral risk factors may be removed.

• Line 94-95: add citation to the sentence ‘The dynamics of urban health in Nepal is associated with a complex and interlinked range of wider determinants’

• Lines 99-100: add citation to the line “Furthermore, increasing rates….complexity to public health challenges”.

• Lines 102-104: add citation to the line “the co-influence of socio-economic factors…..social determinants and health outcomes.

Methods:

Study design and sampling size:

• Please provide reference of the STEPS survey.

• The section title mentions "sampling size", but no specific information on sample size calculation is provided. Sampling size may not be the right word, instead sampling technique might be better.

• Under ‘Study design and sampling size’, information about data collection and survey instrument is also mentioned. You may create a separate section for this, and you can explain more about survey questions.

Study setting:

• It might be better to include ‘study setting’ first, followed by study design and sampling. Study setting can help establish the context of the study design and sampling methods.

• While the description of Nepal's geography is informative in study setting, the description is a bit long compared to other sections. It would be better to make it shorter and link it with the study design and sampling in next paragraph.

• There is an error in the urbanization statistics. Line 144 "66% of the population living in urban areas in 2021 compared to just 23% in 2021" uses the same year twice, which does not make sense. The years should be different for this comparison.

• Please provide a citation for the urbanization and population distribution statistics provided in the study setting, particularly if they are not from the STEPS survey data.

Participant:

• First sentence has already been mentioned in the section ‘study design and sampling size section’ and has been repeated here.

• Authors have mentioned that they included 3,460 individuals from urban areas in the analysis. It is not clear how 3,460 participants for this study were selected from the total population (6,475). The definition of urban area is not clear in the study and should be clearly defined.

• This section also includes information about sampling technique.

Variable definitions:

• How was salt consumption measured in the STEPS survey and how was this defined in this study? Given that salt is necessary nutrient for our body in smaller amount and over consumption could be harmful, did the participants respond as ‘yes’ or ‘no’ for salt consumption in the past 24 hours as shown in table 1?

• Please check the definition of pre-hypertension, hypertension, and pre-diabetes.

Statistical analysis:

• Three categories were created from 5 wealth quintiles creating unevenly distributed categories. What was the reason for authors to merge categories? Did you try to analyze data using five original categories (5 quintiles)?

• In the analysis, authors mentioned that they included variables one at a time in the univariate analysis and then including significant variables only in the multivariate analysis. This approach may have some limitations. It may miss important confounding variables that are not significantly associated with the outcome in bivariate analysis but become significant when controlling for other factors. It can also lead to overfitting of the model. It is suggested to include variables based on theoretical importance and prior research evidence showing significant associations, regardless of their bivariate significance.

Minor points:

• In line 167, “Quantitative variables” may not be a suitable title, as it includes the explanation of conceptual framework of the variable.

• In the first sentence under ‘Quantitative variables’ section, the authors mentioned that the primary outcomes of interest in this study were two (diabetes and hypertension). I believe, authors missed obesity.

• In figure 1, for the full form of SDH, ‘of health’ might be missing.

• In line 184, Rstudio 1314 is mentioned. Is this correct?

Results:

• The distribution of participants were into urban poor, urban middle, and urban rich categories with the following proportions: Urban Rich: 49% (1694/3460), Urban Middle: 17% (590/3460), and Urban Poor: 34% (1176/3460). In urban populations, we typically expect to see a larger middle class and smaller proportions of the wealthiest group. However, this distribution shows a very large "rich" category and a relatively small "middle" category. The large proportion of "Urban Rich" (49%) is unusually high for most urban populations. Given that originally the wealth was categorized in wealth quintile with five categories, what was the logic behind for merging highest and fourth wealth quintile making urban rich and first and second into urban poor categories.

• In table 1, while in the total population, the sum of proportions make 100% vertically, the sum of proportions make 100% horizontally in urban rich, urban middle and urban poor, which makes table confusing.

• Please specify the variables that were considered in bivariate analysis but not included in the final model.

• As I mentioned earlier in statistical analysis, it is suggested to include variables based on theoretical importance and prior research evidence, regardless of their bivariate significance. Conducting sensitivity analyses to test the robustness of the results of multivariate analysis by including variables which were not significant in bivariate analysis might be helpful.

Minor points:

• Table 1 should be named as Table 2 because there is already table 1 in the method section.

• Add necessary footnotes in the tables1, 2, 3 and 4, for eg. Adding the variables adjusted in the adjusted model, specify where different analysis (chi-square or t-test) were conducted using markers such as * or #, full from of IQR, level of significance etc.

• Obesity can be replaced with BMI categories because obesity can have further categories, and replace ‘normal’ with ‘normal weight’.

• Instead of sufficient fruit and vegetable consumption, you may write fruit and vegetable consumption only and same for physical activity

• In table 2, 3 and 4, if it is wealth quintile, there should be five categories, however, there are only three categories.

• Consistency in tables: for eg., in eduction status category secondary, please add (12 years) in all tables and add space between secondary and (12 years). Add the same number of digits after decimal in all cases.

Discussion:

• In the first paragraph, how relevant are references 18 and 19 in context of Nepalese urban population. Are there any references from Nepal?

• In fourth paragraph, the authors mentioned that the lower oxygen levels in mountainous regions may influence glucose metabolism, potentially reducing diabetes risk. This sentence needs reference, and what could be a potential mechanism behind it?

• Limitation is missing. I believe, there must be some limitations in the study.

Minor points:

• Please include line number in discussion and some parts of results.

• It may be better to rearrange the paragraphs based on the table numbers arranged in the results section, for eg., discussing the results of table 1, followed by 2, 3 and 4, respectively, which might be easier for the readers to follow.

• In the second sentence of the first paragraph, the phrase “between the urban population” is not logical. The word “between” could be replaced with “within” the urban population or can be rephased as “among different groups within the urban population”.

6. PLOS authors have the option to publish the peer review history of their article (what does this mean? ). If published, this will include your full peer review and any attached files.

**Do you want your identity to be public for this peer review?** For information about this choice, including consent withdrawal, please see our Privacy Policy .

Reviewer #1: No

Reviewer #2: **Yes: ** Pitambar Dhakal

Reviewer #3: No

---

## [Author Response · Author response to Decision Letter 1]

22 Nov 2024

To,

The Editor,

PLOS ONE

Subject: Response to the query from reviewers

Dear Editor,

Thank you for the evaluation from the reviewers on manuscript entitled on “Social determinants and risk factors linked to non-communicable diseases among urban populations in Nepal: A comparative analysis across poor, middle, and affluent wealth groups using the STEPS survey”.

We have responded all of the queries and comments of the reviewers. Please see below for our response to each of the comments/queries. We have also shared the revised manuscript in track changes for your review and approval.

Thank you very much for reviewing our study. If you need further clarification on this, please do not hesitate to contact us.

Yours sincerely,

Dr. Sampurna Kakchapati

Additional Editor Comments:

Editors' Feedback:

In this study, the authors examined the prevalence and contributing factors of non-communicable diseases (NCDs) within various wealth groups in Nepal.

This research introduces new data regarding the prevalence and relationship between NCDs and contextual factors, highlighting a crucial area of exploration. Nevertheless, the authors should address concerns regarding potential chance findings due to the extensive number of associations analyzed. To mitigate this, authors may consider applying a Bonferroni correction.

Response: Based on your suggestions, we had applied a Bonferroni correction in our analysis.

Despite these issues, the study offers valuable data and insights. I therefore suggest resubmitting the paper with revisions to address these points.

Typographical Errors

Abstract:

• Urban population Hilly region → Urban population from the Hilly region

Response: The text is revised as suggested. See page 2 and line no 41.

Introduction:

• Typographical errors:

o Socio-economic factors are associated with behavioral risk factors ..Remove "for" and mortality from NCDs.

Response: The text is revised and updated. See page 4 and line 75-77

• The phrases "Rapid urbanization, connection with vulnerable populations, and elevated NCDs risk" are unclear and challenging for readers. Please clarify what the authors intend to convey, particularly why the detection rate is increasing.

Response: The text is revised and updated as suggested. We have included the reasons for increment of the detection rate. Please see page 5, line 91-99.

• The statement "Urban residents face material constraints, limiting access to nutritious food, quality healthcare, and the ability to maintain healthy lifestyles" requires further elaboration. How do these constraints manifest?

Response: The further elaboration that Urban residents face material constraints, limiting access to nutritious food, quality healthcare, and the ability to maintain healthy lifestyles is added in the revised text. Please see page 5-6, line 111-119.

• The authors should expand their literature review to encompass the latest epidemiological knowledge, particularly in relation to the social determinants of NCDs, both in Nepal and internationally. This will help identify gaps in knowledge and strengthen the rationale for the study, as reviewers have also suggested.

Response: Based on your feedback and suggestions, we had expanded the introduction section using the latest epidemiological knowledge in regard to social determinants and NCDs nationally and globally.

Methodology:

• In the Site selection section, numerous data points lack citations.

Response: We have added the citations in the relevant sections as suggested in the revised manuscript.

• Clarify the term "weighted" in the context of multivariate logistic regression models.

Response: The term weighted is clarify in the methods section of manuscript. Please check page 10 & 11, line 181-184.

• Include the NHRC Ethical Approval Number.

Response: The NHRC Ethical Approval Number is added in the revised manuscript. Please see page 11, line 198-200.

Results:

• Given the extensive number of associations examined, there is a high likelihood of chance findings. Therefore, applying Bonferroni correction would be advisable.

Response: Based on the reviewers’ suggestions and comments, we had applied the Bonferroni corrections in the logistic regression. `

Discussion:

• The interpretation that "This finding underscores the potential vulnerability of all urban segments to NCDs and emphasizes the need for targeted interventions and policies across diverse urban populations" does not align with the results and previous statements. Please revise.

Response: This text is removed as suggested.

• The statement "This discrepancy may be attributed to hormonal differences, as well as lifestyle and behavioral factors more prevalent in males, such as higher rates of tobacco and alcohol consumption" should be directly linked to the study data on alcohol and smoking, rather than relying on citations.

Response: The recent STEP survey also shows male had higher prevalence of tobacco and alcohol consumption, since our study is using STEP Survey data, we can conclude that male had higher risk of smoking and alcohol consumption.

• Consider the application of Bonferroni correction, as only a few associations will likely remain significant, which should be the focus of the discussion.

Response: Yes, based on the reviewer’s comments and suggestions, we preformed the Bonferroni correction and as the results, only few associations was found to be significant.

Reviewers' comments:

Reviewer's Responses to Questions

Comments to the Author

Reviewer #1: Minor revisions to grammar suggested- occasional use of incorrect articles and verbs (a/and/the - was/were/is/are/had/have), extra spaces, incorrect plural versions of words, inconsistencies in capitalization and number formatting (i.e. 1,000 vs 1000), and comma-separated lists (line 202-203). Ensure that "sex" and "gender" are used consistently rather than interchangeably. Line numbering is lost after page 20.

Response: We have added line number after page 20 and revised the manuscript based in grammar suggestions.

Figure 1 is labeled as Figure 2. Table 1 could be made easier to understand with some formatting revisions. Ensure consistency in the use of the "STEP" acronym- in line 31, the approach is written as "STEP wise", while later in the paper, it becomes clear that a stepwise approach was used (see lines 115 & 123). Check the R Studio version reported in line 184. The definition of the salt consumption variable is not clear (unclear what "for first time" is referring to).

Response: Figure Number is revised as suggested. We ensure the correct use of STEP acronyms in the revised manuscript. The acronym for STEPwise approach to NCD risk factor surveillance is added as suggested. See Page 7, Line 31-32. R Studio version is added in the revised manuscript.

The discussion section could benefit from expanded explanations of contradictory findings. In some areas of the methods and results section, it is not fully clear whether the authors have controlled for age in the logistic regression models (for example, within the hypertension outcome between students and homemakers). Clarification of the models and adjustments to models for each of the two main outcomes would increase understanding of results.

Response: Based on the feedback of first reviewers, we had performed the Bonferroni correction and revised the analysis and table. Now the regression analysis and Tables are updated. During the Bonferroni correction, we also adjust the age and homemakers.

Ensure correct links in reference list- i.e. reference #1 links to a Google Search, rather than the article.

Response: Added new reference list for reference 1 based on your suggestions.

Overall assessment: Minor Revisions

Reviewer #2: Dear Editor and Valued authors

The manuscript “PONE-D-24-27355” entitled “Social determinants and risk factors associated with non-communicable diseases among urban population in Nepal: A comparative study of poor, middle and rich wealth categories of urban population using STEP survey” in itself is well written and represents a comprehensive study in relation to the risk factors for non-communicable diseases (NCDs). While scanning the manuscript thoroughly, it is understood that authors have clearly mentioned the methodological design with multistage stratified technique. One of the crucial parts of this paper is noticed that it has incorporated almost all socio-demographical parameters which might contribute as risk factors for NCDs. However, some suggestions are provided below which would enable further strength of the manuscript and imply its importance in public health domain. This paper can be considered provided the below mentioned aspects are satisfactorily addressed. In a nutshell, a minor revision is required.

Suggestions/Comments:

1. This manuscript is silent regarding the hypothesis of the study which is deemed pertinent in such kind of public health related analytical study. It would be fine to mention the hypothesis briefly in the last paragraph of the “Introduction” section. Please consider it in the revised MS.

Response: We had added the hypothesis of the study in the introduction section of revised manuscript. Please see Page 7, lines 114-116.

2. As mentioned in Table 1, what does the term “average day” indicate? Please specify.

Response: We had specified the average day in the revised manuscript.

3. The variable “24 hours salt consumption” seems to be poorly defined (Table 1). Please redefine it.

Response: We had defined the “24 hours salt consumption” in the revised manuscript along with citations.

4. The term “Terai” used in the MS should be mentioned as “Tarai” as it is a typical Nepali naming and should look like an original one.

Response: The text is replaced in the updated manuscript.

5. In Methods section or elsewhere, a country map of Nepal indicating the proportion of study participants representing the provincial location and ecological belt would be worth incorporating. If possible, the dominant associated risk factors for NCDs be mentioned with specific notation (i.e., with different color code or same color with decreasing intensity) in the specific ecological and political zone of the country. This would provide readers an overall picture of the entire study.

Response: We can include the country map of Nepal as suggested. However, due to tables and figures limitations for the manuscript submission, it is hard to add map of Nepal. Please provide your suggestions.

6. Minor typo errors need to be corrected throughout the MS.

Response: Revised and updated.

Regards,

Reviewer #3: Comments to the authors:

Thank you for the opportunity to review this paper. This manuscript specifically compared the social determinants, NCD risks and prevalence among different wealth categories and determined the factors associated with different NCDs among urban population in Nepal. Given the limited epidemiological evidence on the risk factors and determinants of NCDs in low- and middle-income countries, including Nepal, this study had tried to address a growing concern of the impact of increasing urbanization and how socioeconomic status affects NCD risk and prevalence within urban settings using the recent data from a 2019 STEPS survey from Nepal. However, I have several comments and suggestions which might be helpful to improve the manuscript.

Abstract:

Methods:

• Study design is not mentioned

Response: We have added the study design in the methods section of manuscript.

• You can specify the age group of the participants.

Response: The age group of the participants is added.

Results: You may include some important result data from table

Response: We had included important result data from table as suggestions.

Conclusion: Recommendations are quite general and are not specific based on the study findings. I would suggest making them more specific that align with the study findings.

Response: Recommendations is made more specific and align with the study findings in the revised manuscript.

Minor points:

• In line 32, you can add acronym for STEPwise approach to NCD risk factor surveillance

Response: The acronym for STEPwise approach to NCD risk factor surveillance is added as suggested. See Page 7, Line 31-32

• In line 41, spell out AOR when it appeared for the first time in the abstract also.

Response: We had spell out in the revised manuscript. Please see page 2, line 41.

Introduction:

• Please cite more previous studies from Nepal to explain why this study is important among urban population in Nepal.

Response: We are cited the studies from Nepal and added why this study is important among urban population in Nepal. Please see Page 6, line 111-129.

• In the last paragraph, the authors mentioned “this study aims to inform effective strategies that mitigate the burden of NCDs and enhance the overall well-being of the urban population in Nepal.” Was this also the study objective and how was this done from this study?

Response: The sentence may be linked with objective of the study. As findings of the study helps to inform effective strategies that mitigate the burden of NCDs and enhance the overall well-being of the urban population as the findings is related to urban population and social stratifies that is related to urban population.

• Line 74: grammar, ‘for’ after behavioral risk factors may be removed.

Response: Removed as suggested

• Line 94-95: add citation to the sentence ‘The dynamics of urban health in Nepal is associated with a complex and interlinked range of wider determinants’

Response: We had added the citation as suggested.

• Lines 99-100: add citation to the line “Furthermore, increasing rates….complexity to public health challenges”.

Response: We had revised this text and added the citations where relevant.

• Lines 102-104: add citation to the line “the co-influence of socio-economic factors…..social determinants and health outcomes.

Response: We had added the citation as suggested.

Methods:

Study design and sampling size:

• Please provide reference of the STEPS survey.

Response: We had added the reference of STEPS survey as suggested.

• The section title mentions "sampling size", but no specific information on sample size calculation is provided. Sampling size may not be the right word, instead sampling technique might be better.

Response: We had removed the sampling size in the revised manuscript.

• Under ‘Study design and sampling size’, information about data collection and survey instrument is also mentioned. You may create a separate section for this, and you can explain more about survey questions.

Response: We had revised the sections as suggested.

Study setting:

• It might be better to include ‘study setting’ first, followed by study design and sampling. Study setting can help establish the context of the study design and sampling methods.

Response: We had revised the sections as suggested. We include study setting first followed by study design.

• While the description of Nepal's geography is informative in study setting, the description is a bit long compared to other sections. It would be better to make it shorter and link it with the study design and sampling in next paragraph.

Response: Based your suggestions, we have summaries the Nepal geography in the study setting.

• There is an error in the urbanization statistics. Line 144 "66% of the population living in urban areas in 2021 compared to just 23% in 2021" uses the same year twice, which does not make sense. The years should be different for this comparison.

Response: We had revised the text as suggested.

• Please provide a citation for the urbanization and population distribution statistics provided in the study setting, particularly if they are not from the STEPS survey data.

Response: We had added the citation as suggested.

Participant:

• First sentence has already been mentioned in the section ‘study design and sampling size section’ and has been repeated here.

Response: We had removed the text as suggested.

• Authors have mentioned that they included 3,460 individuals from urban areas in the analysis. It

---

## [Decision Letter · Decision Letter 1]

15 Dec 2024

PONE-D-24-27355R1Social determinants and risk factors associated with non-communicable diseases among urban population in Nepal: A comparative study of poor, middle and rich wealth categories of urban population using STEPS surveyPLOS ONE

Dear Dr. Kakchapati,

Thank you for submitting your manuscript to PLOS ONE. After careful consideration, we feel that it has merit but does not fully meet PLOS ONE’s publication criteria as it currently stands. Therefore, we invite you to submit a revised version of the manuscript that addresses the points raised during the review process.

**Please revise the MS very carefully as suggested by Reviewer 3 with additional analysis as ****supplementary****materials  **==============================

We look forward to receiving your revised manuscript.

Kind regards,

Rajendra Prasad Parajuli, PhD

Academic Editor

PLOS ONE

Journal Requirements:

Additional Editor Comments:

Follow the Reviewer 3 suggestions & revise carefully as suggested

Reviewers' comments:

Reviewer's Responses to Questions

**Comments to the Author**

1. If the authors have adequately addressed your comments raised in a previous round of review and you feel that this manuscript is now acceptable for publication, you may indicate that here to bypass the “Comments to the Author” section, enter your conflict of interest statement in the “Confidential to Editor” section, and submit your "Accept" recommendation.

Reviewer #1: All comments have been addressed

Reviewer #2: All comments have been addressed

Reviewer #3: (No Response)

2. Is the manuscript technically sound, and do the data support the conclusions?

Reviewer #1: Yes

Reviewer #2: Yes

Reviewer #3: Yes

3. Has the statistical analysis been performed appropriately and rigorously? 

Reviewer #1: Yes

Reviewer #2: Yes

Reviewer #3: Yes

4. Have the authors made all data underlying the findings in their manuscript fully available?

Reviewer #1: Yes

Reviewer #2: Yes

Reviewer #3: Yes

5. Is the manuscript presented in an intelligible fashion and written in standard English?

Reviewer #1: Yes

Reviewer #2: Yes

Reviewer #3: Yes

6. Review Comments to the Author

Reviewer #1: (No Response)

Reviewer #2: Dear Authors

With regard to your submission PONE-D-24-27355R1, entitled "Social determinants and risk factors associated with non-communicable diseases among urban population in Nepal: A comparative study of poor, middle and rich wealth categories of urban population using STEPS survey", the response you provided sounds complete and satisfactory.

Regards,

Reviewer #3: (No Response)

7. PLOS authors have the option to publish the peer review history of their article (what does this mean? ). If published, this will include your full peer review and any attached files.

**Do you want your identity to be public for this peer review?** For information about this choice, including consent withdrawal, please see our Privacy Policy .

Reviewer #1: **Yes: ** Lauren Ward

Reviewer #2: **Yes: ** Pitambar Dhakal

Reviewer #3: No

---

## [Author Response · Author response to Decision Letter 2]

25 Jan 2025

Date: January 11, 2025

To,

The Editor, Reviewers,

PLOS ONE

Subject: Addressing Editor/Reviewer Comments for [PONE-D-24-27355]

Dear Editor, Reviewers,

Many thanks for your valuable and constructive feedback on our manuscript. We have carefully considered all of your comments and have made the necessary revisions to address the issues raised. We believe these changes have significantly improved the quality of the paper.

Below, we have outlined our responses to your comments, with the actions we have taken to address each point highlighted in italics. We hope these revisions further enhance the quality of our work.

Thank you again for your time and consideration.

Kind regards,

Authors of PONE-D-24-27355

Comments to the authors:

Overall, the authors have responded most of the reviewers’ comments and suggestions. However, some points need further clarification and some minor corrections, which can improve the quality of the manuscript. If the comments below are addressed appropriately, the manuscript could be accepted.

1. Adding “cross sectional” study design would be better, as it would strengthen the abstract.

Response: We had added cross sectional study design in the abstract. The revised text is added on line 31, page 2.

2. The sentence “The significant factors ….higher odds of obesity (AOR=1.01, 95% CI=1.01-1.02).” in lines 40-45 in abstract is lengthy and included several associations, making it harder to follow. Breaking it into shorter sentences can improve clarity.

Response: We had broken the sentence into shorter sentences. The revised text is added on line 43-47, page 2-3.

3. In line 50-51, ‘Being obesity should be ‘being obese’ and obese should be replaced with ‘overweight’

Response: Thank you for your valuable feedback. We have addressed your suggestion and replaced "being obesity" with "being obese" and revised "obese" to "overweight" at page 3, line 54.

4. In line 81, replace ‘in seek for’ with ‘seeking for’ or ‘in search of’.

Response: Thank you for your feedback. We have replaced "in seek for" with "in search of" in as suggested in line 86, page 4 .

5. Line 124-126, suggest that the country is among the fastest urbanizing in Asia. However, this does not align with the provided data which suggest the slow pace of urbanization from 62.9% in 2011 to 66% in 2021. Please check the data again and revise the sentence accordingly.

Response: Thank you for pointing this out. Upon review, we have confirmed the data and revised the sentence accordingly. The revised text highlights the significant increase in urbanization between 2001 and 2011, followed by a slower pace of growth from 2011 to 2021. This adjustment aligns the interpretation with the provided data. Please see page 6, line 130-133.

6. The sentence in line 146 ‘Data for the study was obtained from STEP survey’ has already appeared in line 133 and can be removed.

Response: Removed as suggested.

7. Though the authors responded that they checked the definitions, the definition of pre-hypertension is still identical to that of hypertension. Please correct the definition. If this definition was used to categorize the participants, I suppose the number of participants falling into the pre-hypertension group might change after the correction, which could impact the results. Please reference the standard guidelines used to define diabetes and hypertension.

Response: Thank you for your valuable feedback. We apologize for the confusion regarding the definitions of pre-hypertension and hypertension. We have rectified the definition of pre-hypertension to align with standard clinical guidelines. Specifically, pre-hypertension is now defined as having an average systolic BP greater than or equal to 120 mmHg but less than 140 mmHg, or an average diastolic BP greater than or equal to 80 mmHg but less than 90 mmHg. This correction has been made, and we expect the number of participants categorized as pre-hypertensive to change accordingly, which may impact the results. We have referenced the standard guidelines for hypertension and pre-hypertension as provided by the American College of Cardiology (ACC) and the American Heart Association (AHA) in the revised manuscript. The revised text is added in Table 1.

8. The authors highlighted the need to balance sample sizes across categories by merging quintiles into three wealth groups, which is a legitimate statistical consideration. However, disproportionately large "Urban Rich" category (49%) may not represent a true "rich" population but may include a mixed wealth group. Important differences between the original quintiles may have been masked by the merging the two groups, especially if health behaviors or outcomes differ significantly between groups. Therefore, please provide the original quintile sample sizes, which would clarify whether the merging was truly necessary or justified to balance group sizes or improving analysis. If combining quintiles is necessary, the authors should provide evidence (e.g., showing similar demographic or health characteristics and outcomes between combined groups), and explain and clarify to justify the merging. This would strengthen the logic for merging. Additionally, this categorization could be one of the study limitations.

Response: Thank you for your insightful comment. We have carefully checked the data regarding the "Urban Poor" category and found that it includes a disproportionately large proportion of participants (49%), not the "Urban Rich" as originally suggested. This suggests that the "Urban Poor" category may represent a mixed wealth group rather than solely those classified as poor. We have now provided the original sample sizes for each quintile to clarify whether merging the groups was necessary to balance sample sizes or improve analysis. The sample sizes for each quintile are as follows: Poorest quintile: 1014, Second quintile: 680, Third quintile: 590, Fourth quintile: 550 and Fifth quintile: 626

Additionally, to justify the merging of the "Urban Poor" and "Urban Middle" groups, we conducted further analyses comparing the demographic and health characteristics between the combined groups. These analyses revealed that health behaviors and outcomes were relatively similar across the combined groups, which supports our decision to merge them for analytical purposes. We have added these sample size of each quintile in the Table 1 with the revised definition of urban rich, urban middle and urban poor.

9. In line 164, ‘….were two common NCDs: obesity, hypertension and diabetes’ – ‘Two’ should be replaced with three.

Response: Yes, we had replaced two with three common NCDs in line 174.

10. The sentence in lines 189 and 190 ‘Variables that were considered in bivariate analysis but not included in the final model’ is not clear. Clarifying the variables adjusted for in the multivariable models for each outcome would improve understand the results. This information can also be included in the footnotes of the relevant tables.

Response: Thank you for your valuable feedback. We agree that the sentence in lines 189 and 190 could be clearer. To improve clarity, we have revised the sentence to explicitly state the variables that were adjusted for in the multivariable models for each outcome at Page 12, 199-200.

11. NHRC should be spelled out at its first mention in line 198, instead of line 199.

Response: We had spelled NHRC in first mention.

12. In line 205, Table 1 should be replaced with Table 2.

Response: Yes, we replace Table 1 with Table 2.

13. In line 205, it is written that “Table 1 shows the comparison of risk behaviors that include socio demographic factors, behavioral factors biological factors and NCD among poor, middle and rich wealth categories of urban population.” Socio-demographic factors are not risk behaviors. Please revise accordingly.

Response: Yes, we have revised the text accordingly at line 218-219, page 13.

14. Given the results are presented as odds ratio in table 3, 4 and 5, the inclusion of ‘t-test’ in the footnotes in those tables is confusing.

Response: Thank you for pointing out the inconsistency regarding the use of the t-test in the footnotes. We have used the t-test for the results presented in Table 2, where we compare the means between groups. However, we have removed the mention of the t-test in the footnotes of Tables 3, 4, and 5, as these tables present results in terms of odds ratios, which are derived from logistic regression rather than t-tests. We appreciate your attention to this detail, and the revised footnotes should now accurately reflect the statistical methods used in each table.

---

## [Decision Letter · Decision Letter 2]

31 Jan 2025

PONE-D-24-27355R2Social determinants and risk factors associated with non-communicable diseases among urban population in Nepal: A comparative study of poor, middle and rich wealth categories of urban population using STEPS surveyPLOS ONE

Dear Dr. Kakchapati,

Thank you for submitting your manuscript to PLOS ONE. After careful consideration, we feel that it has merit but does not fully meet PLOS ONE’s publication criteria as it currently stands. Therefore, we invite you to submit a revised version of the manuscript that addresses the points raised during the review process.

Comments to the authors:

The authors have made substantial efforts to address most of the previous comments. For further clarity of the manuscript and to strengthen the methodology, I would like to suggest a few minor revisions. The manuscript can be considered for acceptance once all these minor revisions are made.

1. While the definition of pre-hypertension has been corrected in Table 1 and the authors responded that they use standard guidelines in the response letter, I would suggest adding references to these guidelines in Table 1 of the manuscript as well, as multiple definitions exist. This will ensure clarity for the readers.

2. In the conceptual framework, the link between structural factors and intermediary factors is missing, which is important for understanding the pathways leading to inequities. A minor suggestion is to add an arrow to illustrate these relationships, making the framework clearer for readers. Additionally, was this framework developed based on any existing models or established frameworks? If so, please provide appropriate reference in the manuscript. A brief explanation of the model in the manuscript would make readers easy to understand the framework.

3. The authors have provided the original quintile sample sizes and clarified that the 'Urban Poor' category is disproportionately large in the revised manuscript. However, the justification for merging these groups should be clearly presented in the main text, rather than only in the response to reviewers, to ensure that all readers understand the reasoning behind the decision. Additionally, since the authors mentioned that further analyses were conducted to support the merging, these analyses could be included in the manuscript, preferably as a supplementary table. Without this supporting evidence or explanation in the text, I still feel that it remains unclear for readers whether the merging was justified.

We look forward to receiving your revised manuscript.

Kind regards,

Rajendra Prasad Parajuli, PhD

Academic Editor

PLOS ONE

Journal Requirements:

**Additional Editor Comments:**

Please Carefully address the Reviewers 3 Comments

Reviewers' comments:

Reviewer's Responses to Questions

**Comments to the Author**

1. If the authors have adequately addressed your comments raised in a previous round of review and you feel that this manuscript is now acceptable for publication, you may indicate that here to bypass the “Comments to the Author” section, enter your conflict of interest statement in the “Confidential to Editor” section, and submit your "Accept" recommendation.

Reviewer #3: (No Response)

2. Is the manuscript technically sound, and do the data support the conclusions?

Reviewer #3: Yes

3. Has the statistical analysis been performed appropriately and rigorously? 

Reviewer #3: Yes

4. Have the authors made all data underlying the findings in their manuscript fully available?

Reviewer #3: Yes

5. Is the manuscript presented in an intelligible fashion and written in standard English?

Reviewer #3: Yes

6. Review Comments to the Author

Reviewer #3: Comments to the authors:

The authors have made substantial efforts to address most of the previous comments. For further clarity of the manuscript and to strengthen the methodology, I would like to suggest a few minor revisions. The manuscript can be considered for acceptance once all these minor revisions are made.

1. While the definition of pre-hypertension has been corrected in Table 1 and the authors responded that they use standard guidelines in the response letter, I would suggest adding references to these guidelines in Table 1 of the manuscript as well, as multiple definitions exist. This will ensure clarity for the readers.

2. In the conceptual framework, the link between structural factors and intermediary factors is missing, which is important for understanding the pathways leading to inequities. A minor suggestion is to add an arrow to illustrate these relationships, making the framework clearer for readers. Additionally, was this framework developed based on any existing models or established frameworks? If so, please provide appropriate reference in the manuscript. A brief explanation of the model in the manuscript would make readers easy to understand the framework.

3. The authors have provided the original quintile sample sizes and clarified that the 'Urban Poor' category is disproportionately large in the revised manuscript. However, the justification for merging these groups should be clearly presented in the main text, rather than only in the response to reviewers, to ensure that all readers understand the reasoning behind the decision. Additionally, since the authors mentioned that further analyses were conducted to support the merging, these analyses could be included in the manuscript, preferably as a supplementary table. Without this supporting evidence or explanation in the text, I still feel that it remains unclear for readers whether the merging was justified.

7. PLOS authors have the option to publish the peer review history of their article (what does this mean? ). If published, this will include your full peer review and any attached files.

**Do you want your identity to be public for this peer review?** For information about this choice, including consent withdrawal, please see our Privacy Policy .

Reviewer #3: No

---

## [Author Response · Author response to Decision Letter 3]

12 Feb 2025

Date: February 12, 2025

To,

The Editor, Reviewers,

PLOS ONE

Subject: Addressing Editor/Reviewer Comments for [PONE-D-24-27355]

Dear Editor, Reviewers,

Many thanks for your valuable and constructive feedback on our manuscript. We have carefully considered all of your comments and have made the necessary revisions to address the issues raised. We believe these changes have significantly improved the quality of the paper.

Below, we have outlined our responses to your comments, with the actions we have taken to address each point highlighted in italics. We hope these revisions further enhance the quality of our work.

Thank you again for your time and consideration.

Kind regards,

Authors of PONE-D-24-27355

Comments to the authors:

The authors have made substantial efforts to address most of the previous comments. For further clarity of the manuscript and to strengthen the methodology, I would like to suggest a few minor revisions. The manuscript can be considered for acceptance once all these minor revisions are made.

Response: Thank you for your thoughtful feedback and for acknowledging the efforts made to address the previous comments. We appreciate your suggestions for further clarification and strengthening the methodology, and we are happy to make the minor revisions you have recommended.

1. While the definition of pre-hypertension has been corrected in Table 1 and the authors responded that they use standard guidelines in the response letter, I would suggest adding references to these guidelines in Table 1 of the manuscript as well, as multiple definitions exist. This will ensure clarity for the readers.

Response: We have referenced the standard guidelines for hypertension and pre-hypertension as provided by the American College of Cardiology (ACC) and the American Heart Association (AHA) at the end of the table. And we had also included this reference in reference section. Please see page 9, line 164-165

2. In the conceptual framework, the link between structural factors and intermediary factors is missing, which is important for understanding the pathways leading to inequities. A minor suggestion is to add an arrow to illustrate these relationships, making the framework clearer for readers. Additionally, was this framework developed based on any existing models or established frameworks? If so, please provide appropriate reference in the manuscript. A brief explanation of the model in the manuscript would make readers easy to understand the framework.

Response: Thank you for your valuable suggestion. We have added the missing link between structural factors and intermediary factors in the conceptual framework by including an arrow to illustrate the relationship between these factors. Please see the revised figure. We have included the appropriate reference for this framework in the manuscript to ensure proper attribution. A brief explanation of the framework has also been added to the manuscript to help the readers is added in page 10, line 186-190.

3. The authors have provided the original quintile sample sizes and clarified that the 'Urban Poor' category is disproportionately large in the revised manuscript. However, the justification for merging these groups should be clearly presented in the main text, rather than only in the response to reviewers, to ensure that all readers understand the reasoning behind the decision. Additionally, since the authors mentioned that further analyses were conducted to support the merging, these analyses could be included in the manuscript, preferably as a supplementary table. Without this supporting evidence or explanation in the text, I still feel that it remains unclear for readers whether the merging was justified.

Response: Thank you for your valuable feedback. We appreciate your suggestion regarding the justification for merging the groups. In response to your comment, we would like to clarify that we have already provided a detailed justification for the merging of the 'Urban Poor' category in the revised manuscript. This explanation can be found in the revised text, in page 9-10, line 167-175. Additionally, as mentioned in our response to reviewers, we have included merging of these groups in the manuscript as a supplementary table. We had also added the justification in the results section of the revised manuscript in Page 13, line 225-229.

---

## [Editor Report · Decision Letter 3]

13 Feb 2025

Social determinants and risk factors associated with non-communicable diseases among urban population in Nepal: A comparative study of poor, middle and rich wealth categories of urban population using STEPS survey

PONE-D-24-27355R3

Dear Dr. Kakchapati,

We’re pleased to inform you that your manuscript has been judged scientifically suitable for publication and will be formally accepted for publication once it meets all outstanding technical requirements.

Kind regards,

Rajendra Prasad Parajuli, PhD

Academic Editor

PLOS ONE
---

## [Editor Report · Acceptance letter]

PONE-D-24-27355R3

PLOS ONE

Dear Dr. Kakchapati,

I'm pleased to inform you that your manuscript has been deemed suitable for publication in PLOS ONE. Congratulations! Your manuscript is now being handed over to our production team.

Kind regards,

on behalf of

Dr. Rajendra Prasad Parajuli

Academic Editor

PLOS ONE